# Exploration by Running Away from the Past

## Abstract

The ability to explore efficiently and effectively is a central challenge of reinforcement learning. In this work, we consider exploration through the lens of information theory. Specifically, we cast exploration as a problem of maximizing the Shannon entropy of the state occupation measure. This is done by maximizing a sequence of divergences between distributions representing an agent's past behavior and its current behavior. Intuitively, this encourages the agent to explore new behaviors that are distinct from past behaviors. Hence, we call our method RAMP, for "**R**unning **A**way fro**m** the **P**ast." A fundamental question of this method is the quantification of the distribution change over time. We consider both the Kullback-Leibler divergence and the Wasserstein distance to quantify divergence between successive state occupation measures, and explain why the former might lead to undesirable exploratory behaviors in some tasks. We demonstrate that by encouraging the agent to explore by actively distancing itself from past experiences, it can effectively explore mazes and a wide range of behaviors on robotic manipulation and locomotion tasks.

## 1 Introduction

Exploration is essential in reinforcement learning (RL) as it allows agents to discover optimal strategies in complex environments. Without it, agents risk becoming stuck in suboptimal policies, lacking the diverse experiences needed to learn effectively. Despite a long history of study, with fundamental algorithms like R-max (Brafman & Tennenholtz, 2002), UCRL (Auer & Ortner, 2006), and E3 (Kearns & Singh, 2002), exploration remains a major challenge in modern RL.

One method for encouraging exploration is providing intrinsic motivation to the agent; our approach falls into this category. The agent aims to maximize this additional (intrinsic) reward which motivates exploration. Various intrinsic reward models specific to exploration have been developed (Brafman & Tennenholtz, 2002; Auer & Ortner, 2006; Bellemare et al., 2016; Eysenbach et al., 2018; Badia et al., 2020). Amongst these are methods based on the principle of *optimism in the face of uncertainty* (Munos et al., 2014), where agents are encouraged to explore under-visited areas of the state space by assigning high reward values to uncertain states. While these methods apply well to tabular state spaces, in high-dimensional or continuous state spaces, these methods must rely on parameterized functions to represent uncertainty, which may result in inconsistent behavior (Pathak et al., 2017; Burda et al., 2018).

Information theory provides a useful perspective on exploration. For example, one can define a finite set of skills which have some descriptor per skill. Maximizing the Mutual Information (MI) between the descriptors of these skills, and the states which they visit, yields good exploratory behaviors for this set of skills (Eysenbach et al., 2018). Similarly, it has been demonstrated that maximizing the Shannon entropy of the state distribution encourages exploration (Liu & Abbeel, 2021; Hazan et al., 2019; Lee et al., 2019). However, these approaches often rely on probability density estimators, and finding a relevant density estimator for an environment can be challenging.

This study introduces a method for exploration that aims at achieving a high Shannon entropy score of the distribution representing the agent's experiences. We show that this objective can be reframed as the maximization of a sequence of Kullback-Leibler (KL) divergences between successive state occupation measures. This new objective leads to the intuitive use of a simple classifier as a density estimator. In other words, the goal of exploration, represented by high Shannon entropy over the agent's experiences, can be achieved by iteratively separating an agent's past experiences with its

most recent ones. This results in an algorithm called RAMP, for "**R**unning **A**way fro**m** the **P**ast," as the agent explores by distancing itself from its past experiences.

We evaluate the performance of RAMP using multiple metrics, including the state space coverage and the maximum score achieved by a policy trained to maximize a specific reward model. In environments where exploration could benefit from a metric relevant to the state space, we suggest using the Wasserstein distance as an alternative to the KL divergence. We study the difference between these two measures by proposing two versions of the algorithm: $\text{RAMP}_{\text{KL}}$, which uses KL divergence, and $\text{RAMP}_{\mathcal{W}}$, which uses Wasserstein distance. We compare RAMP to several baselines developed for exploration tasks and across various environments such as mazes, locomotion tasks, and robotics tasks. We find that RAMP is competitive in exploration to state-of-the-art methods such as LSD (Park et al., 2022) on robotics tasks, and that it leads to more effective exploration on maze and locomotion tasks. We conclude that this reformulation of the Shannon entropy objective can open further avenues in the domain of exploration in RL. All our implementations are available at the following repository: GitHub repository.

## 2 PROBLEM STATEMENT

We first formally define the objectives of the RAMP algorithm. Let us consider a reward-free Markov decision process (Puterman, 2014, MDP) $(\mathcal{S}, \mathcal{A}, P, \delta_0)$ where $\mathcal{S}$ is the state space, $\mathcal{A}$ the action space, $P$ the transition function and $\delta_0$ the initial state distribution. A behavior policy $\pi_{\theta_n}(s)$ parameterized by $\theta_n$ maps states to distributions over actions. Here, $n$ corresponds to an epoch in the policy optimization sequence. Given a fixed horizon $T$, the average occupation time, or state occupancy measure density induced by $\pi_{\theta_n} = \pi_n$ over the state space is:

$$\rho_n(s) = \mathbb{E}_{\substack{s_1 \sim \delta_0 \\ a_t \sim \pi_n(\cdot|s_t) \\ s_{t+1} \sim P(\cdot|s_t, a_t)}} \left[ \frac{1}{T} \sum_{t=1}^{T} \mathbb{1}_s(s_t) \right]$$

Note that the work presented herein applies seamlessly when $T$ tends to infinity. Given a parameter $\beta \in (0, 1)$, let $\mu_n$ be the $(1 - \beta)$-discounted mixture of past state occupancies, up to epoch $n$:

$$\mu_n(s) = \beta \sum_{k=1}^{n} (1 - \beta)^{n-k} \rho_k(s) \qquad \mu_{n+1}(s) = \beta \underbrace{\rho_{n+1}(s)}_{\text{The present}} + (1 - \beta) \underbrace{\mu_n(s)}_{\text{The past}}$$

Consider epoch $n + 1$, $\rho_{n+1}$ denotes the agent's current occupancy measure density, representing its present behavior, while $\mu_n$ reflects its past experience. In practice, a sample of $\mu_n$ can be maintained using a replay buffer. At each epoch, the replay buffer is updated by retaining a proportion $(1 - \beta)$ from the previous buffer and incorporating a proportion $\beta$ from the new distribution. Intuitively, $(1 - \beta)$ specifies the extent to which past distributions are retained in $\mu_n$. It should be noted that this definition differs from typical practice in Deep Reinforcement Learning, where it is generally assumed that all data is stored in the replay buffer (Mnih et al., 2013).

We write $H_n = H_{\mu_n}[S]$, the Shannon entropy of distribution $\mu_n$. Optimizing the diversity of occupied states at epoch $n$, by maximizing $H_n$, is an appealing objective for promoting exploration up to epoch $n$. To this end, one can ensure a monotonic increase of $H_n$ throughout the epochs:

$$H_{n+1} - H_n > 0 \qquad \forall n \in \mathbb{N} \tag{1}$$

Specifically $\Delta_{n+1} = H_{n+1} - H_n$ is the entropy increase rate. Noting $D_{\text{KL}}$ as the Kullback-Leibler divergence, $\Delta_{n+1}$ admits the following lower bound (proof in Appendix B).

**Theorem 1** (Lower Bound on $\Delta_{n+1}$)**.**

$$\Delta_{n+1} \geq \beta \left( D_{KL}(\rho_{n+1} || \mu_{n+1}) + H_{\rho_{n+1}}[S] - H_n \right)$$

This lower bound provides an optimization objective for $\pi_{n+1}$ through the proxy of $\rho_{n+1}$. Specifically, at epoch $n + 1$, $\mu_n$ is the mixture of past occupancy measures. To ensure a positive entropy increase rate, one searches for $\rho_{n+1}$ such that $D_{\text{KL}}(\rho_{n+1} || \underbrace{\beta\rho_{n+1} + (1 - \beta)\mu_n}_{\mu_{n+1}}) + H_{\rho_{n+1}}[S] \geq H_n$.

A classic proxy in RL for obtaining a large state occupation $\rho^\pi$ for a given policy $\pi$ (and hence, a large $H_{\rho^\pi}[S]$), is the maximization of the policy's entropy on average across encountered states $\mathbb{E}_{s \sim \rho^\pi}[H_{\rho^\pi}[A|S]]$. The rationale is that taking random actions may induce a widespead state distribution. This hypothesis may not universally apply across all environments, but empirical findings presented in this study, as well as the vast literature on the benefits of entropy regularization for exploration (Haarnoja et al., 2018; Geist et al., 2019; Ahmed et al., 2019), provide robust evidence supporting its practical applicability. Therefore, we take take $\pi_{n+1}$ such that:

$$\pi_{n+1} = \underset{\pi}{\operatorname{argmax}} \; \underbrace{\underset{s \sim \rho^\pi}{\mathbb{E}} \left[ \log \left( \frac{\rho^\pi(s)}{\beta \rho^\pi + (1-\beta)\mu_n(s)} \right) \right]}_{\text{repulsive term}} + \lambda_A \underset{\substack{s \sim \rho^\pi \\ a \sim \pi(\cdot|s)}}{\mathbb{E}} \left[ -\log(\pi(a|s)) \right], \qquad (2)$$

with $\lambda_A > 0$. By maximizing the KL divergence (the repulsive term) in Equation 2, our goal is to generate a distribution $\rho_{n+1}$ that significantly deviates from the mixture of prior distributions. In other words, we wish to explore by "running away from the past". Appendix H discusses how this term is related to the divergence between $\rho^\pi$ and $\mu_n$.

However, the quality of the exploration achieved in practice by maximizing the KL divergence in Equation 2 may vary across contexts. For example, in high-dimensional continuous state spaces such as Ant (Brockman et al., 2016), maximizing this divergence can be trivially done by manipulating the agent's joints, therefore visiting new configurations of the agent, without the need to explore new locomotion behaviors.

The limitation arises from the fact that KL divergence is not sensitive to the underlying geometry of the state space, treating all state changes equivalently without regard to the spatial distance between states. To address this, it can be advantageous to use a divergence that takes the metric structure of the state space into account. The Wasserstein distance (Villani et al., 2009), also known as the Earth Mover's Distance, provides a principled way of measuring differences between distributions by considering the cost of transporting mass between states, thereby encouraging exploration that covers distinct regions of the environment. The Wasserstein distance is in itself an optimization problem. In our case, it can be defined through the Kantorovich duality as:

$$\mathcal{W}(\rho^\pi, \beta \rho^\pi + (1-\beta)\mu_n) = \max_{\|f\| \le 1} \; \underset{s^+ \sim \rho^\pi}{\mathbb{E}} \left[ f(s^+) \right] - \underset{s^- \sim \beta \rho^\pi + (1-\beta)\mu_n}{\mathbb{E}} \left[ f(s^-) \right], \qquad (3)$$

where $f$ is a 1-Lipshitz function for a given metric $d$ defined over $\mathcal{S}$: $\forall (s_1, s_2) \in \mathcal{S}^2 \; \| \; f(s_2) - f(s_1) \; \| \le d(s_1, s_2)$. In this work, we use the Temporal Distance $d^{\text{temp}}(s_1, s_2)$ (Kaelbling, 1993; Hartikainen et al., 2019; Durugkar et al., 2021; Park et al., 2023b), which represents the minimum number of steps that must be performed in a Markov chain in order to reach state $s_1$ from $s_2$. When using the Wasserstein distance instead of the Kullback-Leibler divergence, we take $\pi_{n+1}$ such that:

$$\pi_{n+1} = \underset{\pi}{\operatorname{argmax}} \; \underbrace{\mathcal{W}(\rho^\pi, \beta \rho^\pi + (1-\beta)\mu_n)}_{\text{repulsive term}} + \lambda_A \underset{\substack{s \sim \rho^\pi \\ a \sim \pi(\cdot|s)}}{\mathbb{E}} \left[ -\log(\pi(a|s)) \right]. \qquad (4)$$

Note that this objective does not maximize a lower bound on $\Delta_n$ per se. However it remains a well-motivated formulation to promote exploration. Overall, the objective of achieving high Shannon entropy over experiences naturally leads to the maximization of a divergence between past experiences and recent ones. We make this our primary objective in the remainder of the paper, either in the form of Equation 2 or 4. We use this framing to encourage exploration by defining the corresponding intrinsic motivation rewards.

## 3 RUNNING AWAY FROM THE PAST (RAMP)

The core idea of RAMP is to explore by running away from the past. We now define how we use intrinsic motivation to reward this movement, which leads to the above objectives of maximizing entropy. The RAMP algorithm uses two alternative estimates of distribution divergence, $r_{D_{KL}}(s)$ and $r_{\mathcal{W}}(s)$, to reward the agent for moving away from past experiences.

### 3.1 INTRINSIC REWARD MODELS

Objectives 2 and 4 incorporate repulsive terms that maximize specific divergences between the distributions $\rho^\pi$ and $\beta \rho^\pi + (1-\beta)\mu_n$. Let us define $r_{D_{KL}}^\pi(s) = \log \left( \rho^\pi(s) / (\beta \rho^\pi(s) + (1-\beta)\mu_n(s)) \right)$

the reward model based on the first term in Equation 2. This term can thus be written $\langle \rho^\pi, r^\pi_{D_{KL}} \rangle$. Suppose one has access to an estimate $\hat{r}_{D_{KL}}(s)$ of $r^\pi_{D_{KL}}(s)$. Finally, let us define $\pi'$ as a policy that is better or equivalent than $\pi$ for the reward model $\hat{r}_{D_{KL}}$. Then one can prove that the divergence between $\rho^{\pi'}$ and $\rho^{\pi'}\beta + (1-\beta)\mu_n$ is larger than that for $\pi$. In other terms, $\pi'$ runs further away from the past than $\pi$. Formally, this yields (proof in Appendix C):

**Theorem 2.** *Given policy $\pi$, let $\varepsilon_1$ be the approximation error of $\hat{r}_{D_{KL}}$, i.e. $\|\hat{r}_{D_{KL}} - r^\pi_{D_{KL}}\|_\infty \leq \varepsilon_1$.*

*Let $\pi'$ be another policy and $\varepsilon_0 \in \mathbb{R}$ such that $\|\frac{\rho^{\pi'}}{\rho^\pi} - 1\|_\infty \geq \varepsilon_0$ ($\rho^{\pi'}$ is close to $\rho^\pi$).*

*Finally, let $\varepsilon_2$ measure how much $\pi'$ improves on $\pi$ for $\hat{r}_{D_{KL}}$: $\langle \rho^{\pi'}, \hat{r}_{D_{KL}} \rangle - \langle \rho^\pi, \hat{r}_{D_{KL}} \rangle = \varepsilon_2$.*

*If $\varepsilon_2 \geq 2\varepsilon_1 - \log(1 - \varepsilon_0)$, then $D_{KL}\left(\rho^{\pi'}||\rho^{\pi'}\beta + (1-\beta)\mu_n\right) \geq D_{KL}\left(\rho^\pi||\rho^\pi\beta + (1-\beta)\mu_n\right)$.*

For the repulsive term in Objective 4, the reward model is defined as $r_{\mathcal{W}}(s) = f^*(s)$, where $f^*$ belongs to the solutions of the problem defined in Equation 3. Theorem 3 states that maximizing the reward model $r_{\mathcal{W}}$ leads to the maximization of the Wasserstein distance (proof in Appendix D).

**Theorem 3.** *Given policy $\pi$, let $\varepsilon_1$ be the approximation error of $\hat{r}_{\mathcal{W}}$, i.e. $\|\hat{r}_{\mathcal{W}} - r^\pi_{\mathcal{W}}\|_\infty \leq \varepsilon_1$.*

*Let $\pi'$ be another policy and $\varepsilon_2$ measure how much $\pi'$ improves on $\pi$ for $\hat{r}_{\mathcal{W}}$: $\langle \rho^{\pi'}, \hat{r}_{\mathcal{W}} \rangle - \langle \rho^\pi, \hat{r}_{\mathcal{W}} \rangle = \varepsilon_2$. If $\varepsilon_2 \geq 2\varepsilon_1(1+\beta)$, then $\mathcal{W}(\rho^{\pi'}, \beta\rho^{\pi'} + \mu_n(\beta - 1)) > \mathcal{W}(\rho^\pi, \beta\rho^\pi + \mu_n(\beta - 1))$.*

This poses a critical question: how to estimate $r_{D_{KL}}(s)$ and $r_{\mathcal{W}}(s)$? We start with the simpler of the two, the estimation of $r_{D_{KL}}$.

## 3.2 ESTIMATING $r_{D_{KL}}$

As proposed by Eysenbach et al. (2020), estimating the log of the ratio between two different distributions can be seen as a contrastive learning problem. Consider a neural network with parameters $\phi$, $f_\phi : \mathcal{S} \rightarrow \mathbb{R}$, and the following labeling:

$$\begin{cases} s^+ \sim \rho^\pi \iff L = 1 \\ s^- \sim \beta\rho^\pi + (1-\beta)\mu_n \iff L = 0 \end{cases} \tag{5}$$

To solve this classification problem, one can minimize the following loss:

$$\mathcal{L}_{D_{KL}}(\phi) = - \mathop{\mathbb{E}}_{\substack{s^+ \sim \rho^\pi \\ s^- \sim \beta\rho^\pi + (1-\beta)\mu_n}} \left[ \log\left(\sigma(f_\phi(s^+))\right) + \log\left(1 - \sigma(f_\phi(s^-))\right) \right] \tag{6}$$

When the proportions of positive and negative samples are the same (the probability of label 0, $P(L = 0)$, is the same as the probability of label 1, $P(L = 1)$), Bayes' rule gives:

$$P(L = 1|s) = \frac{P(s|L=1)}{P(s|L=0) + P(s|L=1)} = \frac{\rho^\pi(s)}{\rho^\pi(s) + \beta\rho^\pi(s) + (1-\beta)\mu_n(s)}$$

Using the sigmoid function $\sigma(f_\phi(s))$ to regress the labels, we obtain:

$$\frac{1}{1 + e^{-f_\phi(s)}} \approx \frac{\rho^\pi(s)}{\rho^\pi(s) + \beta\rho^\pi(s) + (1-\beta)\mu_n(s)} \Leftrightarrow f_\phi(s) \approx \log\left(\frac{\rho^\pi(s)}{\beta\rho^\pi(s) + (1-\beta)\mu_n(s)}\right)$$

Therefore, by solving this simple classification problem, the output of the neural network (without the sigmoid activation) is exactly $r_{D_{KL}}$.

## 3.3 ESTIMATING $r_{\mathcal{W}}$

To estimate a solution of the Wasserstein distance between two distributions, the temporal distance can be used (Durugkar et al., 2021; Park et al., 2023b). A solution $f^*$ to the Wasserstein optimization problem defined in equation 3 can be approximated by a neural network $f_\phi$, using dual gradient descent with a Lagrange multiplier $\lambda$ and a small relaxation constant $\epsilon$. As proposed by Durugkar et al. (2021), the 1-Lipschitz constraint under the temporal distance is maintained by ensuring that:

$$\sup_{s \in \mathcal{S}} \left\{ \mathbb{E}_{s' \sim P(\cdot|s,a)} \left[ |f(s) - f(s')| \right] \right\} \leq 1$$

This is done by minimizing the following loss with SGD:

$$\mathcal{L}_{\mathcal{W}}(\lambda, \phi) = - \mathop{\mathbb{E}}_{s^+ \sim \rho^\pi} \left[ f(s^+) \right] + \mathop{\mathbb{E}}_{s^- \sim \beta \rho^\pi + (1-\beta)\mu_n} \left[ f(s^-) \right]$$

$$- \lambda \cdot \mathop{\mathbb{E}}_{\substack{s \sim (1+\beta)\rho^\pi + (1-\beta)\mu_n \\ a \sim \pi(\cdot|s) \\ s' \sim P(\cdot|s,a)}} \left( \max \left( |f_\phi(s) - f_\phi(s')| - 1, -\epsilon \right) \right) \quad (7)$$

In practice, the optimization involves taking gradient steps on $\lambda$ followed by gradient steps on $\phi$. Here, $\lambda$ is adjusted adaptively to weight the constraint, taking high values when the constraint is violated and low (but positive) values when it is satisfied. The resulting neural network $f_\phi$ corresponds to the reward model $r_{\mathcal{W}}$.

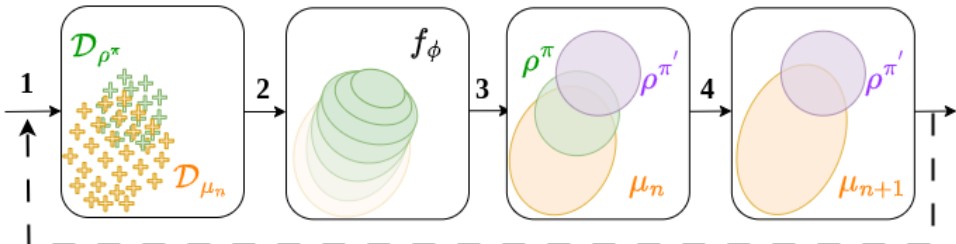

Figure 1: The four steps of the RAMP algorithm.

### 3.4 The RAMP Algorithm

The full RAMP algorithm is described in Algorithm 1 and illustrated in Figure 1. The RAMP algorithm follows four key steps. First, current policy $\pi$ is used to **(1) sample new experiences in the environment** to measure the policy $\pi$'s current occupancy measure. In practice, a buffer $\mathcal{D}_\rho$ containing $N_\rho^e$ episodes from the agent's most recent experiences is used.

Next, the instrinsic reward model is updated to better **(2) estimate the new versus past experiences**. In RAMP, the reward model is represented by a simple neural network $f_\phi$ which is the solution to a specific optimization problem. Two alternate measures of divergence are possible: the KL divergence and the Wasserstein distance. Thus, the RAMP algorithm has two versions: $\text{RAMP}_{\text{KL}}$, which maximizes Objective 2, and $\text{RAMP}_{\mathcal{W}}$, which maximizes Objective 4. For $\text{RAMP}_{\text{KL}}$, $f_\phi$ is determined by solving the contrastive problem in Equation 6. Alternatively, for $\text{RAMP}_{\mathcal{W}}$, $f_\phi$ is obtained by solving the constrained optimization problem described in Equation 7.

The third step **(3) maximizes the difference between the present and the past** distributions. The reward models proposed by RAMP can be maximized using any RL method. This study uses the Soft Actor-Critic (SAC) algorithm Haarnoja et al. (2018) for all experiments. The final step is to **(4) update the distribution of past experiences** $\mu_n$. In practice, only a sample of $\mu_n$ contained in a buffer $\mathcal{D}_{\mu_n}$ is available. The goal is to transform this sample throughout learning so that its empirical distribution mirrors $\mu_n$ at each epoch. The full details of constructing $\mathcal{D}_{\mu_n}$ are given in Appendix F.

We study the RAMP algorithm on a series of control tasks, focusing on exploration of robotic locomotion in the following sections. We compare RAMP to contemporary methods for exploration, which are described below.

## 4 Related Work

The use of intrinsic rewards to explore generic state spaces has been extensively studied. In this section, we review the corresponding literature, starting with standard exploration bonuses defined in tabular cases, and progressing to more complex exploration bonuses using deep neural networks.

**Tabular case.** In the tabular case, the most straightforward way to explore a state space effectively is by defining a reward that is inversely proportional to the number of times $n_s = \sum_{t=1}^N \mathbb{1}(s_t = s)$

---

**Algorithm 1:** RAMP$_{(\text{KL}, \mathcal{W})}$

---

**Input:** $\beta, N, N_\rho^e, \lambda_A, \lambda$

$\pi_{\theta_0}, f_\phi, \mathcal{D}_{\mu_0} = \{s \sim \tau_{\pi_{\theta_0}}\}, \mathcal{D}_\rho = \{\}$                    // Initialization

**for** *epoch* $\leftarrow 1$ **to** $N + 1$ **do**

    $\mathcal{D}_\rho = \{\}$                    // Reset present experience buffer

    **for** *episode* $\leftarrow 1$ **to** $N_\rho^e$ **do**

        $\tau = \{s_t, a_t, s_{t+1}\}_{t \in [0,T]}$  // Sample environment with current policy

        $\mathcal{D}_\rho = \mathcal{D}_\rho \bigcup \tau$                    // Update empirical estimation of $\rho^\pi$

    $\phi = \arg\max_\phi \mathcal{L}_{\text{D}_{\text{KL}}}(\phi)$ or $\phi, \lambda = \arg\max_\phi \mathcal{L}_{\mathcal{W}}(\phi, \lambda)$

    Obtain $\pi_{\theta_{n+1}}$ using SAC to maximize: $E_{s \sim \rho}[f_\phi(s)]$

    $\mathcal{D}_{\mu_{n+1}} = \{s \sim \mathcal{D}_\rho \quad \text{if} \quad A \sim \mathcal{B}(\beta) \quad \text{else} \quad s \sim \mathcal{D}_{\mu_n}\}$                    // Update past

---

a state has been encountered during the $N$ steps of training. This exploration strategy has been used in methods such as E3 (Kearns & Singh, 2002), R-max Brafman & Tennenholtz (2002) or UCRL Auer & Ortner (2006); Jaksch et al. (2010); Bourel et al. (2020) to improve the theoretical bounds for the algorithm's online performance after a finite number of steps.

**Uncertainty measure on generic state spaces.** In general, precisely estimating how often an agent visits a state is not always feasible, especially in continuous or high-dimensional state spaces. To address this, various methods have been proposed to approximate state visit counts. Bellemare et al. (2016) introduced a density-based method to estimate visit counts recursively. More recent approaches define a task-specific loss function, $f_\phi(s) = l(\phi, s)$, as a proxy for $\frac{1}{n_s}$, using the loss as an intrinsic reward to guide exploration. For example, Shelhamer et al. (2016) and Jaderberg et al. (2016) use the VAE loss, while Pathak et al. (2017) proposed the Intrinsic Curiosity Module (ICM), which uses prediction error as an intrinsic reward. To address vanishing rewards, Burda et al. (2018) introduced Random Network Distillation (RND), and Badia et al. (2020) extended this with Never Give-Up (NGU), which adds a trajectory-based bonus for continued exploration even when uncertainty decreases.

**Information Theory based algorithms.** In Hazan et al. (2019), exploration is posed as the maximization of the Shanon Entropy $H_\mu[S] = \mathbb{E}_{s \sim \mu}[-\log(\mu(s))]$ defined for the replay buffer distribution $\mu$. They show that optimizing a policy for the reward model $r(s) = -\log(\mu(s))$ leads to state occupation entropy maximization. Liu & Abbeel (2021) proposed an unsupervised active pretraining (APT) method that learns a representation of the state space, which is subsequently used by a particle-based density estimator aimed at maximizing $H_\mu[S]$. Eysenbach et al. (2018) introduced the notion of skill-based exploration where the policy is conditioned by a skill descriptor, and propose a method to maximize the Mutual Information (MI) between states visited by skills and their descriptors. The objective of skill-based exploration methods (Gregor et al., 2016; Sharma et al., 2019; Campos et al., 2020; Kamienny et al., 2021) is to derive a policy that leads to very different and distinguishable behaviors, when conditioned with different realizations of $Z$. This concept was then hybridized with an uncertainty measure by Lee et al. (2019) to tackle the exploration limitations of MI based methods.

**Mixing Information Theory and metrics.** Information theory offers tools to compare distributions, typically without considering the underlying metric. However, in continuous or high-dimensional spaces, using a relevant metric can improve exploration efficiency. Park et al. (2022) and Park et al. (2023a) exploit this by combining information theory with Euclidean distance, maximizing the Wasserstein distance version of mutual information using the Euclidean distance as the state space metric. Recently, Park et al. (2023b) furthered this approach, still maximizing the Wasserstein distance, but using the temporal distance.

**Compared strengths and weaknesses.** Each exploration method has its own advantages and limitations. Uncertainty-based methods like ICM (Pathak et al., 2017) and RND (Burda et al., 2018) frame exploration as an adversarial game where the policy maximizes an uncertainty-based reward,

while the uncertainty estimator minimizes the estimation error. However, finding an equilibrium in this game can be numerically challenging. Entropy maximization methods, such as APT (Liu & Abbeel, 2021), rely on computationally expensive density estimators with weak theoretical guarantees. Additionally, approaches combining mutual information with a metric often require many samples due to the joint optimization of the agent and a discriminator.

RAMP takes a different approach by incrementally improving the Shannon entropy of the agent's state distribution over time, rather than directly maximizing it. This avoids the costly density estimators used in particle-based methods (Liu & Abbeel, 2021; Badia et al., 2020). Unlike skill-diversity algorithms (Eysenbach et al., 2018; Park et al., 2023b), RAMP does not require generalizing across behaviors, which may result in more sample-efficient exploration. By focusing on incremental improvement, RAMP offers a promising solution for efficient exploration.

## 5 EXPERIMENTS AND RESULTS

In this section, we study the capacity of the RAMP algorithm to lead to exploration in complex environments. We begin with a qualitative analysis of the reward models used by RAMP to better characterize the objectives of RAMP. We then perform a quantitative analysis of the exploration obtained by RAMP, using coverage of the state space as the exploration metric. Finally, we analyze the ability of RAMP to explore when also using an additional, extrinsic, reward.

The evaluation is conducted on three different sets of tasks, which are illustrated and further detailed in Appendix I. The first is maze navigation, for which there are three custom mazes of varying difficulty. Secondly, we use five robotic locomotion environments from the MuJoCo platform (Todorov et al., 2012): Ant, HalfCheetah, Hopper, Humanoid, and Walker2d. Finally, we study exploration in robot control tasks from the Gymnasium platform (de Lazcano et al., 2023): Fetch Reach, Fetch Push, and Fetch Slide.

### 5.1 UNDERSTANDING RAMP'S REWARD MODELS

We first aim to illustrate how RAMP constructs consecutive coverage distributions to encourage exploration. Figure 2a shows the positional information of a robot in the U-maze environment at a given training epoch. The color gradient corresponds to the density estimated by the classifier of $\text{RAMP}_{\text{KL}}$ for the states contained in $\mathcal{D}_{\mu_n}$ (the brown shape) and $\mathcal{D}_\rho$ (the green shape), illustrating that the classifier learns to distinguish the distributions of past states $\mu_n$ and present states $\rho^\pi$. The reward produced by the classifier encourages the agent to explore new areas by moving away from the past experiences contained in $\mathcal{D}_{\mu_n}$. In this low-dimensional state space, the KL divergence gives a good approximation of the difference in distributions, so we can expect that $\text{RAMP}_{\text{KL}}$ will motivate the exploration of new states.

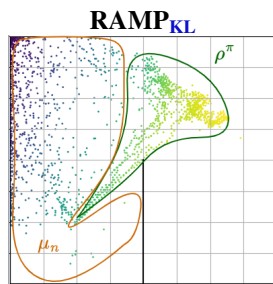 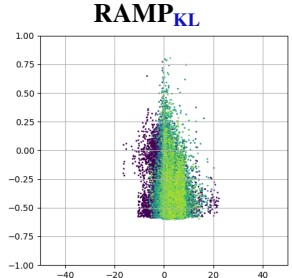 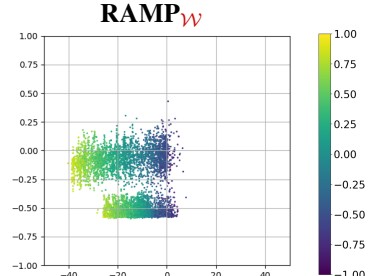

(a) $XY$-coordinates of the agent in the U-maze.

(b) $YZ$-coordinates of the HalfCheetah's torso. The color indicates the density used as the reward model for $\text{RAMP}_{\text{KL}}$ and $\text{RAMP}_{\mathcal{W}}$.

Figure 2: (a) An illustration of the different experience distributions on the U-maze. (b) A comparison between $\text{RAMP}_{\text{KL}}$ and $\text{RAMP}_{\mathcal{W}}$ on HalfCheetah. Color indicates the reward estimate given by $f_\phi$, normalized between -1 and 1.

However, as discussed in Section 2, maximizing the KL divergence between $\mu_n$ and $\rho^\pi$ in high-dimensional state space can lead to limited exploration for specific tasks. Figure 2b shows the

exploration performed by RAMP$_{\text{KL}}$ (left) compared to RAMP$_{\mathcal{W}}$ (right) in the HalfCheetah environment. In this environment, the agent's state space has a dimensionality of 18, including information on the various robot limbs, so the classifier $f_\phi$ may focus on states related to joint configuration rather than the overall position. We note this in the left image, where the reward estimated by the classifier for RAMP$_{\text{KL}}$ does not sufficiently encourage the agent to explore new $YZ$-coordinates. In the right image, the reward model $f_\phi$ for RAMP$_{\mathcal{W}}$ attributes higher reward for states where the robot's center is further away from the origin. The use of the Wasserstein distance, instead of the KL divergence, encourages meaningful exploration in this case.

Figure 3 demonstrates that the exploration of RAMP$_{\mathcal{W}}$ is effective even in higher-dimensional spaces. This figure depicts a timeline representing the $XY$-coordinates of the torso of the Ant robot at different points in training. This version of the Ant robot simulator has 113 variables in each state, including nonessential information like the forces applied to various joints. The agent quickly learns to move away from the initial distribution, which has a mean located at $(X = 0, Y = 0)$. The agent then learns to distance itself from its past experiences by circling the environment, ultimately resulting in a uniform distribution over the $XY$ coordinates. Despite the many possible combinations of variables in the high-dimensional state of the Ant robot, RAMP$_{\mathcal{W}}$ is able to reward exploration that creates new movements of behavior, beyond simple exploration of new states.

**RAMP$_{\mathcal{W}}$ exploration in Ant**

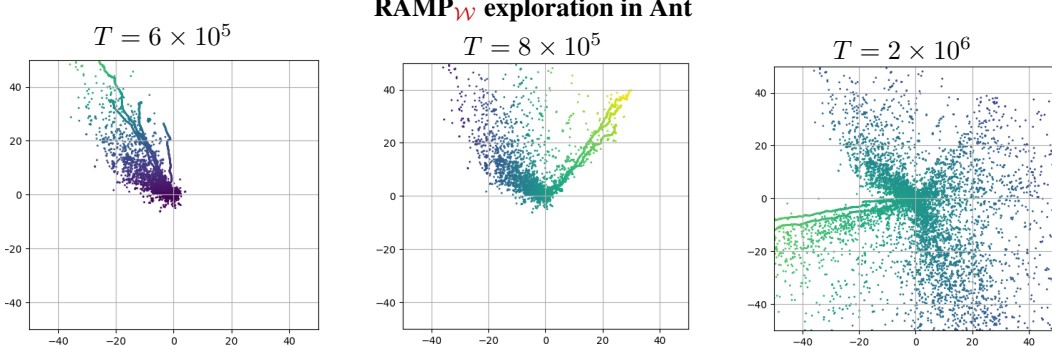

Figure 3: $XY$-coordinates of the Ant's torso at different timesteps $T$ of training. Color indicates the density used as reward model for RAMP$_{\mathcal{W}}$.

## 5.2 QUANTIFYING THE CAPACITY TO EXPLORE

We now compare the two versions of RAMP (RAMP$_{\text{KL}}$ and RAMP$_{\mathcal{W}}$) with 10 baselines which represent different approaches to exploration. We aim to understand whether the objective of RAMP leads to effective exploration, compared to similar and contemporary methods of exploration, and how the two methods of reward estimation in RAMP compare across environments.

We compare with state-of-the-art baselines which encourage exploration through the use of uncertainty and information theory. For approaches based on **uncertainty**, we compare with AUX (Jaderberg et al., 2016), ICM, (Pathak et al., 2017), RND (Burda et al., 2018), and NGU (Badia et al., 2020). For approaches based on **information theory**, we compare with APT (Liu & Abbeel, 2021), DIAYN (Eysenbach et al., 2018), SMM (Lee et al., 2019), LSD (Park et al., 2022), METRA (Park et al., 2023b). Soft Actor Critic (SAC) (Haarnoja et al., 2018) is used as the default reward maximizer for each of these methods and is also included in the comparison. For skill-based algorithms, such as DIAYN, we use a fixed number of 4 skills for all environments.

For each method, we calculate the state space coverage after a fixed number of environment steps. State space coverage is computed by discretizing the agent's state space in a low-dimensional space, which is specific to each environment, and which is initialized with zero values. During training, each time a new state is encountered, the corresponding matrix index is updated to one. The coverage corresponds to the percentage of the matrix that is filled. This way of quantifying exploration through state space coverage has wide use in the domain of Quality Diversity algorithms (Pugh et al., 2016). The coverage is assessed on the $xy$-coordinates for the mazes, the $xyz$-coordinates for the locomotion tasks and the Fetch Reach environment, and the $xy$-coordinates of the objects for Fetch Slide and Fetch Push. For each of the baselines, the final coverage is evaluated after $5 \times 10^5$ steps

for the mazes, $1 \times 10^6$ steps for the robotics tasks, and $8 \times 10^6$ steps for the locomotion tasks. The results for the locomotion tasks are shown in Table 1. Each column of the table indicates the relative mean coverage obtained by each method, divided by the coverage achieved by the best runs across all baselines for that environment. Complementary results of state space coverage on the maze and robotics tasks are detailed in Appendix A.

| Algorithm | Ant | HalfCheetah | Hopper | Humanoid | Walker2d |
|---|---|---|---|---|---|
| APT | 7.68 ± 0.9 | **93.39 ± 3.39** | 55.52 ± 3.75 | 54.73 ± 2.97 | 55.37 ± 2.83 |
| AUX | 4.53 ± 0.03 | 18.87 ± 0.21 | 5.65 ± 0.16 | 58.2 ± 0.24 | 11.65 ± 0.5 |
| DIAYN | 11.76 ± 0.61 | 58.18 ± 5.65 | 15.03 ± 8.89 | 70.68 ± 4.11 | 14.84 ± 1.34 |
| ICM | 3.26 ± 0.13 | 28.9 ± 1.55 | 41.63 ± 0.56 | 58.96 ± 0.87 | 33.81 ± 1.95 |
| LSD | 7.01 ± 2.15 | 30.43 ± 2.79 | 18.38 ± 5.27 | 69.89 ± 2.25 | 17.26 ± 2.32 |
| METRA | 23.46 ± 0.74 | 73.82 ± 3.0 | 37.5 ± 3.33 | 88.35 ± 5.05 | 36.88 ± 4.18 |
| NGU | 2.79 ± 0.07 | 25.53 ± 0.42 | 19.55 ± 0.62 | 44.02 ± 4.3 | 27.26 ± 2.01 |
| RND | 4.57 ± 0.07 | 19.1 ± 0.14 | 6.95 ± 0.51 | 57.91 ± 0.2 | 13.19 ± 0.31 |
| SAC | 4.4 ± 0.05 | 18.42 ± 0.24 | 5.83 ± 0.15 | 57.94 ± 0.27 | 13.11 ± 0.92 |
| SMM | 10.61 ± 1.28 | 58.91 ± 5.2 | 43.41 ± 14.93 | 32.64 ± 3.09 | 44.21 ± 13.19 |
| RAMP$_{\text{KL}}$ | 1.2 ± 0.05 | 29.76 ± 1.12 | 8.6 ± 0.55 | 30.53 ± 1.54 | 17.52 ± 2.04 |
| RAMP$_{\mathcal{W}}$ | **78.35 ± 13.45** | 40.33 ± 6.69 | **74.43 ± 12.14** | **90.5 ± 2.29** | **74.89 ± 11.71** |

Table 1: Final relative mean coverage for the robot locomotion tasks. Bold indicates the highest mean per environment.

RAMP$_{\mathcal{W}}$ outperforms all baseline methods in state space coverage across all but one locomotion task, HalfCheetah. The most notable performance is on the Ant environment, where RAMP$_{\mathcal{W}}$ achieves a final score that is six times higher than that of the second-best baseline. As postulated above, RAMP$_{\text{KL}}$ does not perform effective exploration on these environments, given the high dimensionality of the state space. However, RAMP$_{\text{KL}}$ does provide a good reward estimate for exploration in simpler environments, as shown on the mazes in Appendix A, and when an extrinsic reward is provided, as discussed below.

The limitations of RAMP are demonstrated on the robotic control tasks and detailed in Appendix A. While RAMP is able to explore effectively on robot locomotion tasks, the control tasks such as FetchPush and FetchSlide are challenging for RAMP. While RAMP achieves competitive results to contemporary methods such as DIAYN (Eysenbach et al., 2018) and NGU (Badia et al., 2020), APT (Liu & Abbeel, 2021) significantly outperforms RAMP. We posit that a different density estimator, such as the one used in APT, could improve RAMP's performance, and that the presence of extrinsic motivation on these tasks would further help in exploration.

## 5.3 Exploring with extrinsic reward

We next evaluate the capacity of RAMP to aid in the maximization of an extrinsic reward. For example, in the locomotion tasks, the robots are rewarded for moving away from their starting point. In this experiment, the reward maximized is a weighted sum of the intrinsic reward, from RAMP, and the extrinsic rewards provided by the environment. To enable the agent to focus more on the extrinsic reward as training progresses, the weight of the intrinsic reward decreases linearly over time, reaching zero halfway through the training process. In this experiment, the two versions of RAMP are compared with a subset of the above baseline methods; a comparison with skill-based algorithms goes beyond the scope of the study, as these methods are often used in a hierarchical setting. All methods are run for $4 \times 10^4$ timesteps in each environment, using the same extrinsic and intrinsic motivation weighting, where applicable. The results are presented in Table 2, which displays the mean maximum score achieved by each policy across five independent trials.

Both variations of RAMP demonstrate strong performance. As in pure exploration, the most significant outcome is observed in the Ant environment, where RAMP$_{\mathcal{W}}$ exceeds the second-best baseline by over 40%. This result was unexpected, given that the exploration strategy employed by RAMP$_{\mathcal{W}}$

| Algorithm | Ant | HalfCheetah | Hopper | Humanoid | Walker2d |
|---|---|---|---|---|---|
| APT | 1042 ± 69 | 10316 ± 138 | 3036 ± 442 | 3330 ± 739 | 2205 ± 313 |
| AUX | 5434 ± 263 | 11127 ± 243 | 2249 ± 465 | 3477 ± 706 | 4767 ± 91 |
| ICM | 4450 ± 402 | 11161 ± 163 | 3675 ± 126 | 3880 ± 491 | 5513 ± 191 |
| NGU | 975 ± 12 | 2976 ± 584 | 1360 ± 60 | 415 ± 93 | 1689 ± 96 |
| RND | 4427 ± 158 | 10901 ± 108 | 3084 ± 381 | 5103 ± 34 | 5723 ± 56 |
| SAC | 4972 ± 95 | 12197 ± 79 | **3875 ± 40** | 5163 ± 70 | 5650 ± 108 |
| RAMP$_{KL}$ | 4768 ± 381 | **13826 ± 361** | 3636 ± 206 | **5358 ± 49** | **5939 ± 524** |
| RAMP$_{\mathcal{W}}$ | **7100 ± 47** | 12997 ± 987 | 1036 ± 67 | 5342 ± 74 | 5933 ± 174 |

Table 2: Cumulative episodic return for the robot locomotion tasks. Bold indicates the highest mean per environment.

does not appear to be optimal for maximizing performance on locomotion tasks, which necessitate precise joint synchronization. We hypothesized that the success of RAMP$_{\mathcal{W}}$ can be attributed to the fact that, in the initial learning phase, the intrinsic reward may provide a more manageable optimization signal, enabling the agent to learn rapid movement, thus facilitating the subsequent optimization of the extrinsic reward, locomotion behaviors.

Interestingly, RAMP$_{KL}$, slightly outperforms RAMP$_{\mathcal{W}}$ on average, with aggregate mean scores across the five environments of $33,527 \times 10^3$ and $32,408 \times 10^3$, respectively. We posit that RAMP$_{KL}$ is able to effectively explore environments, even when the state space is large, if the exploration is conditioned towards novel behaviors. Given that the extrinsic reward signal encourages locomotion, the intrinsic motivation given by KL divergence will focus on the difference in state variables like the robot position, rather than simply iterating over various robot joint configurations. As RAMP$_{KL}$ is both simpler to implement and more closely linked theoretically to the original goal of maximizing Shannon entropy over visited states, we consider that the KL divergence is a logical choice when exploration can be guided towards interesting behaviors. RAMP$_{\mathcal{W}}$, on the other hand, is able to explore effectively in the presence or absence of extrinsic reward.

## 6 CONCLUSION

This paper presents a novel exploration method in reinforcement learning aimed at maximizing the Shannon entropy of an agent's experience distribution. By reformulating this objective, we derive a new method, RAMP, that encourages the agent to explore by distancing itself from past experiences.

We investigate the use of KL divergence and Wasserstein distance to characterize the differences between the agent's current distribution and its past distribution. We characterize two versions of RAMP, (RAMP$_{\mathcal{W}}$ and RAMP$_{KL}$), and compare them. We find that maximizing the Wasserstein distance between two distributions under the temporal distance intuitively results in a different behavior compared to maximizing the KL divergence. To maximize the Wassterein distance, the agent must encounter states that are as far away as possible from the states already encountered. This leads to effective exploration and even the ability to reach high rewards.

Our evaluations reveal that, in locomotion tasks, RAMP$_{\mathcal{W}}$ enables highly efficient exploration by extensively exploring over the agent's position. In the presence of extrinsic reward, RAMP$_{KL}$ also demonstrates the ability to explore sufficiently to lead to rewarding behaviors. Beyond the use of an extrinsic reward, RAMP$_{KL}$ could also be performed by measuring the KL divergence of a featurization of the state space, rather than the full state. This could lead to exploration of new behaviors, as demonstrated under the presence of an extrinsic reward. We aim to study such a featurization for high-dimensional state spaces in future work.

In summary, this work reframes exploration in reinforcement learning as an iterative process of distinguishing the agent's present behavior from its past experiences, introducing a novel approach with wide applicability. The RAMP algorithm offers a new framing of exploration that could be combined with other exploration strategies, such as skill-based or hierarchical exploration. This new perspective on exploration, driven by maximizing the divergence between successive distributions, has the potential to advance both theoretical insights and practical algorithms for exploration in reinforcement learning.

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

## A ADDITIONAL EXPERIMENTS

### A.1 COVERAGE

Table 3 contains the results of the relative coverage for the point maze environments in Figure 7. $RAMP_{KL}$ and $RAMP_{\mathcal{W}}$ demonstrate effective performance across all maze environments, with

| Algorithm | Easy-maze | Hard-maze | U-maze |
|---|---|---|---|
| APT | 98.39 ± 0.31 | **77.54 ± 0.86** | 89.47 ± 1.8 |
| AUX | 26.52 ± 0.32 | 26.96 ± 0.2 | 26.53 ± 0.16 |
| DIAYN | 54.5 ± 2.86 | 40.5 ± 1.93 | 45.37 ± 6.42 |
| ICM | 83.83 ± 2.82 | 66.45 ± 11.48 | 67.15 ± 6.87 |
| LSD | 74.93 ± 2.13 | 46.01 ± 1.99 | 52.24 ± 2.46 |
| METRA | 53.47 ± 4.99 | 38.71 ± 5.74 | 51.09 ± 3.57 |
| NGU | 58.95 ± 1.88 | 42.32 ± 4.41 | 44.01 ± 5.36 |
| RND | 27.07 ± 0.22 | 27.26 ± 0.11 | 26.9 ± 0.17 |
| SAC | 27.41 ± 0.23 | 27.65 ± 0.25 | 27.06 ± 0.28 |
| SMM | 56.3 ± 3.17 | 42.94 ± 1.25 | 48.15 ± 5.53 |
| $RAMP_{KL}$ | 98.51 ± 0.44 | 71.23 ± 4.08 | 79.56 ± 2.24 |
| $RAMP_{\mathcal{W}}$ | **100.0 ± 0.0** | 70.47 ± 0.6 | **89.5 ± 3.14** |

Table 3: Final coverage in mazes

$RAMP_{\mathcal{W}}$ achieving the highest coverage in two specific tasks. Although $RAMP_{\mathcal{W}}$ exhibits slightly better performance than $RAMP_{KL}$ in these environments, the difference is not substantial.

The limitations of RAMP become apparent in robotic tasks, as illustrated by the coverage results presented in Table 4. In this context, neither of the RAMP methods attains the highest coverage in one of the tasks. While $RAMP_{\mathcal{W}}$ achieves the second highest coverage in the FetchReach environment, it is notably outperformed by APT (Liu & Abbeel, 2021) in environments where coverage is assessed on the object to be manipulated. This disparity arises because the intrinsic reward mechanism derived from the state representation employed by APT provides a highly effective learning signal, enabling the agent to achieve significantly competitive coverage in a highly sample-efficient manner. Interestingly, APT significantly outperform $RAMP_{KL}$ while these two methods essentially maximize the same objective in different ways and using different density estimators. This further emphasizes the importance of the density estimator used to address a given task.

| Algorithm | FetchPush | FetchReach | FetchSlide |
|---|---|---|---|
| APT | **90.87 ± 3.59** | 59.37 ± 1.92 | **95.43 ± 4.5** |
| AUX | 20.27 ± 0.91 | 19.5 ± 0.09 | 46.59 ± 2.55 |
| DIAYN | 18.12 ± 0.54 | 33.98 ± 2.24 | 45.05 ± 3.68 |
| ICM | 36.32 ± 4.24 | 77.51 ± 1.16 | 42.68 ± 2.21 |
| LSD | 36.24 ± 6.01 | **98.08 ± 0.72** | 47.49 ± 1.75 |
| METRA | 10.99 ± 0.43 | 68.05 ± 1.38 | 17.01 ± 1.47 |
| NGU | 23.85 ± 0.94 | 62.22 ± 1.22 | 44.91 ± 1.99 |
| RND | 21.86 ± 0.32 | 19.49 ± 0.16 | 54.66 ± 2.57 |
| SAC | 21.24 ± 0.73 | 19.41 ± 0.1 | 47.14 ± 2.7 |
| SMM | 69.06 ± 5.98 | 29.37 ± 2.29 | 64.96 ± 1.26 |
| $RAMP_{KL}$ | 23.61 ± 3.15 | 58.57 ± 2.5 | 35.55 ± 2.83 |
| $RAMP_{\mathcal{W}}$ | 34.2 ± 6.18 | 81.56 ± 1.42 | 51.11 ± 2.9 |

Table 4: Final coverage in robotic tasks

## A.2 EPISODIC RETURN

Table 5 depicts the mean maximum score reached by the best policy obtained using the various algorithms.

| Algorithm | Easy-maze | Hard-maze | U-maze |
|---|---|---|---|
| APT | **1.27 ± 0.0** | 0.56 ± 0.05 | **1.0 ± 0.0** |
| AUX | **1.27 ± 0.0** | 0.51 ± 0.0 | 0.5 ± 0.0 |
| ICM | **1.27 ± 0.0** | 0.61 ± 0.1 | 0.8 ± 0.12 |
| RND | **1.27 ± 0.0** | 0.51 ± 0.0 | 0.9 ± 0.1 |
| SAC | 1.03 ± 0.01 | 0.51 ± 0.0 | 0.5 ± 0.0 |
| RAMP$_{KL}$ | 1.25 ± 0.02 | 0.51 ± 0.0 | 0.64 ± 0.1 |
| RAMP$_{\mathcal{W}}$ | **1.27 ± 0.0** | **0.69 ± 0.15** | 0.9 ± 0.1 |

Table 5: Cumulative episodic return in mazes

The results were obtained after $2.5 \times 10^5$ steps in the environment. Once again, the exploration conducted by RAMP$_{\mathcal{W}}$ enables the agent to achieve the highest return in two mazes and the second highest return in the U-maze. While these environments are relatively simple, this experiment can be interpreted as a sanity check. Notably, in all environments, the baselines utilizing intrinsic rewards for exploration consistently achieved scores equal to or higher than those of Vanilla SAC.

## B PROOF OF THEOREM 1

For convenience, we recall Theorem 1.

**Theorem.** *Given two successive epochs $n$ and $n+1$, the rate of increase $\Delta_n$ of the Shannon entropy of the $\mu_n$ distribution can be lower bounded as follows:*

$$H_{n+1} - H_n \geq \beta \left( D_{KL}(\rho_{n+1} || \mu_{n+1}) + H_{\rho_{n+1}}[S] - H_n \right)$$

*Proof.* By definition:

$$H_{n+1} - H_n = \int_{\mathcal{S}} \mu_{n+1}(s) \log \left( \frac{1}{\mu_{n+1}(s)} \right) + \mu_n(s) \log (\mu_n(s)) \, \mathrm{d}s$$

By definition of $\mu_{n+1}$:

$$H_{n+1} - H_n =$$

$$= \int_{\mathcal{S}} \beta \rho_{n+1}(s) \log \left( \frac{1}{\mu_{n+1}(s)} \right) + (1-\beta) \mu_n(s) \log \left( \frac{1}{\mu_{n+1}(s)} \right) + \mu_n(s) \log (\mu_n(s)) \, \mathrm{d}s$$

$$= \int_{\mathcal{S}} \beta \rho_{n+1}(s) \log \left( \frac{\rho_{n+1}(s)}{\mu_{n+1}(s)} \right) + (1-\beta) \mu_n(s) \log \left( \frac{\mu_n(s)}{\mu_{n+1}(s)} \right) +$$

$$\beta \mu_n(s) \log (\mu_n(s)) + \beta \rho_{n+1}(s) \log \left( \frac{1}{\rho_{n+1}(s)} \right) \, \mathrm{d}s$$

$$= \beta \left( D_{KL}(\rho_{n+1} || \mu_{n+1}) + H_{\rho_{n+1}}[S] - H_n \right) + (1-\beta) D_{KL}(\mu_n || \mu_{n+1})$$

KL is always a positive quantity: $\forall n \in \mathbb{N}^*$: $D_{KL}(\mu_n || \mu_{n+1}) \geq 0$. Therefore, the entropy's speed of increase can be lower bounded the following way:

$$H_{n+1} - H_n \geq \beta \left( D_{KL}(\rho_{n+1} || \mu_{n+1}) + H_{\rho_{n+1}}[S] - H_n \right)$$

This ends the proof. □

**Discussion:** By definition $D_{KL}(\mu_n || \mu_{n+1})$ is evaluated on the support of $\mu_n$. whereas when maximizing the entropy's rate of increase one could expect a KL defined on $\rho_{n+1}$'s support (or $\mu_{n+1}$'s support) to be much more informative. This study focuses on the left term.

## C    PROOF OF THEOREM 2

For the sake of convenience, we recall Theorem 2.

**Theorem.** *Given policy $\pi$, let $\varepsilon_1$ be the approximation error of $\hat{r}_{D_{KL}}$, i.e. $\|\hat{r}_{D_{KL}} - r^\pi_{D_{KL}}\|_\infty \leq \varepsilon_1$.*

*Let $\pi'$ be another policy and $\varepsilon_0 \in \mathbb{R}$ such that $\|\frac{\rho^{\pi'}}{\rho^\pi} - 1\|_\infty \geq \varepsilon_0$ ($\rho^{\pi'}$ is close to $\rho^\pi$).*

*Finally, let $\varepsilon_2$ measure how much $\pi'$ improves on $\pi$ for $\hat{r}_{D_{KL}}$: $\langle \rho^{\pi'}, \hat{r}_{D_{KL}} \rangle - \langle \rho^\pi, \hat{r}_{D_{KL}} \rangle = \varepsilon_2$.*

*If $\varepsilon_2 \geq 2\varepsilon_1 - \log(1 - \varepsilon_0)$, then $D_{KL}\left(\rho^{\pi'} || \rho^{\pi'}\beta + (1-\beta)\mu_n\right) \geq D_{KL}\left(\rho^\pi || \rho^\pi\beta + (1-\beta)\mu_n\right)$.*

*Proof.* For the sake of notation simplicity, we write $\rho' = \rho^{\pi'}$ and $\rho = \rho^\pi$. We also introduce the function $f(x,y) = \log(x/(\beta x + (1-\beta)y))$. Note that, for a fixed $y$, $f$ is monotonically increasing in $x$.

Finally, given any real-valued functions $\nu$ and $\mu$, we note $f(\nu, \mu)$ the function $s \mapsto f(\nu(s), \mu(s))$.

Then:

$$
\begin{aligned}
D_{KL}\left(\rho' || \rho'\beta + (1-\beta)\mu_n\right) &= \mathop{\mathbb{E}}_{s \sim \rho'}\left[\log\left(\frac{\rho'(s)}{\beta\rho'(s) + (1-\beta)\mu_n(s)}\right)\right], \\
&= \mathop{\mathbb{E}}_{s \sim \rho'}\left[f(\rho', \mu_n)(s)\right], \\
&= \langle \rho', f(\rho', \mu_n) \rangle.
\end{aligned}
$$

Using the assumption that $\rho'$ is close to $\rho$, one has $1 - \varepsilon_0 \leq \frac{\rho'(s)}{\rho(s)} \leq 1 + \varepsilon_0$ for all $s$. Thus $\rho'(s) \geq \rho(s)(1 - \varepsilon_0)$, and:

$$
\begin{aligned}
\langle \rho', f(\rho', \mu_n) \rangle &\geq \langle \rho', f((1-\varepsilon_0)\rho, \mu_n), \\
&\geq \mathop{\mathbb{E}}_{s \sim \rho'}\left[\log\left(\frac{\rho(s)(1-\varepsilon_0)}{\beta(\rho(s))(1-\varepsilon_0) + (1-\beta)\mu_n(s)}\right)\right], \\
&\geq \mathop{\mathbb{E}}_{s \sim \rho^{\pi'}}\left[\log\left(\frac{\rho(s)}{\beta(\rho(s))(1-\varepsilon_0) + (1-\beta)\mu_n(s)}\right)\right] + \log((1-\varepsilon_0)).
\end{aligned}
$$

Additionally, $\log\left(\frac{\rho(s)}{\beta(\rho(s))(1-\varepsilon_0)+(1-\beta)\mu_n(s)}\right) \geq \log\left(\frac{\rho(s)}{\beta(\rho(s))+(1-\beta)\mu_n(s)}\right)$ and so:

$$
\begin{aligned}
\langle \rho', f(\rho', \mu_n) \rangle &\geq \mathop{\mathbb{E}}_{s \sim \rho'}\left[\log\left(\frac{\rho(s)}{\beta(\rho(s)) + (1-\beta)\mu_n(s)}\right)\right] + \log((1-\varepsilon_0)), \\
&\geq \langle \rho', f(\rho, \mu_n) \rangle + \log((1-\varepsilon_0)).
\end{aligned}
$$

Recall that $\|\hat{r}_{D_{KL}} - r^\pi_{D_{KL}}\|_\infty \leq \varepsilon_1$, that is $|\hat{r}_{D_{KL}}(s) - r^\pi_{D_{KL}}(s)| \leq \varepsilon_1$ for all $s$. Using the $f$ notation: $|\hat{r}_{D_{KL}}(s) - f(\rho, \mu_n)(s)| \leq \varepsilon_1$. And hence $f(\rho, \mu_n)(s) \geq \hat{r}_{D_{KL}}(s) - \varepsilon_1$ for all $s$. Consequently:

$$
\langle \rho', f(\rho, \mu_n) \rangle \geq \langle \rho', \hat{r}_{D_{KL}} \rangle - \varepsilon_1.
$$

And so:

$$
\langle \rho', f(\rho', \mu_n) \rangle \geq \langle \rho', \hat{r}_{D_{KL}} \rangle + \log(1 - \varepsilon_0) - \varepsilon_1.
$$

By definition of $\varepsilon_2$, one has $\langle \rho', \hat{r}_{D_{KL}} \rangle - \langle \rho, \hat{r}_{D_{KL}} \rangle = \varepsilon_2$. So:

$$
\langle \rho', f(\rho', \mu_n) \rangle \geq \langle \rho, \hat{r}_{D_{KL}} \rangle + \log(1 - \varepsilon_0) - \varepsilon_1 + \varepsilon_2.
$$

As previously, since $\|\hat{r}_{D_{KL}} - r^\pi_{D_{KL}}\|_\infty \leq \varepsilon_1$, one has $|\hat{r}_{D_{KL}}(s) - f(\rho, \mu_n)(s)| \leq \varepsilon_1$. And hence $\hat{r}_{D_{KL}}(s) \geq f(\rho, \mu_n)(s) - \varepsilon_1$. Therefore:

$$
\langle \rho, \hat{r}_{D_{KL}} \rangle \geq \langle \rho, f(\rho, \mu_n) \rangle \geq -\varepsilon_1.
$$

Putting it all together, one obtains:

$$
\langle \rho', f(\rho', \mu_n) \rangle \geq \langle \rho, f(\rho, \mu_n) \rangle + \log(1 - \varepsilon_0) - 2\varepsilon_1 + \varepsilon_2.
$$

We identify $D_{KL}\left(\rho' || \rho'\beta + (1-\beta)\mu_n\right)$ in the first term and $D_{KL}\left(\rho || \rho\beta + (1-\beta)\mu_n\right)$ in the second one.

Finally, if $\log(1 - \varepsilon_0) - 2\varepsilon_1 + \varepsilon_2 \geq 0$ we guarantee that the first divergence is larger than the second, which concludes the proof. $\qquad\square$

## D  PROOF OF THEOREM 3

For convenience, we recall Theorem 3.

**Theorem.** *Given policy $\pi$, let $\varepsilon_1$ be the approximation error of $\hat{r}_{\mathcal{W}}$, i.e. $\|\hat{r}_{\mathcal{W}} - r_{\mathcal{W}}^\pi\|_\infty \leq \varepsilon_1$. Let $\pi'$ be another policy and $\varepsilon_2$ measure how much $\pi'$ improves on $\pi$ for $\hat{r}_{\mathcal{W}}$: $\langle \rho^{\pi'}, \hat{r}_{\mathcal{W}} \rangle - \langle \rho^\pi, \hat{r}_{\mathcal{W}} \rangle = \varepsilon_2$. If $\varepsilon_2 \geq 2\varepsilon_1(1 + \beta)$, then $\mathcal{W}(\rho^{\pi'}, \beta\rho^{\pi'} + \mu_n(\beta - 1)) > \mathcal{W}(\rho^\pi, \beta\rho^\pi + \mu_n(\beta - 1))$.*

*Proof.* By definition :

$$
\mathcal{W}(\rho^\pi, \beta\rho^\pi + \mu_n(\beta - 1)) = \max_{\substack{f \\ \text{s.t.} \\ \|f\| \leq 1}} \mathbb{E}_{s \sim \rho^\pi}[f(s)] - \mathbb{E}_{s \sim \rho^\pi \beta + (1-\beta)\mu_n}[f(s)]
$$

$$
= \max_{\substack{f \\ \text{s.t.} \\ \|f\| \leq 1}} \langle \rho^\pi, f \rangle - \langle (\beta\rho^\pi + (1-\beta)\mu_n), f \rangle
$$

$$
= \max_{\substack{f \\ \text{s.t.} \\ \|f\| \leq 1}} (1 + \beta)\langle \rho^\pi, f \rangle - (1 - \beta)\langle \mu_n, f \rangle
$$

We wish to define a lower bound of this quantity which incorporates $\hat{r}_{\mathcal{W}}$ so that we can later use the definition of $\varepsilon_2$. By definition, $r_{\mathcal{W}}^\pi$ is one solution $f*$ of this maximization problem and one can write :

$$
\mathcal{W}(\rho^\pi, \beta\rho^\pi + \mu_n(\beta - 1)) = (1 + \beta)\langle \rho^\pi, r_{\mathcal{W}}^\pi \rangle - (1 - \beta)\langle \mu_n, r_{\mathcal{W}}^\pi \rangle
$$

Recall that $\|\hat{r}_{\mathcal{W}} - r_{\mathcal{W}}^\pi\|_\infty \leq \varepsilon_1$, that is $|\hat{r}_{\mathcal{W}}(s) - r_{\mathcal{W}}^\pi(s)| \leq \varepsilon_1$ for all $s$. So we have two inequalities $-\varepsilon_1 \leq \hat{r}_{\mathcal{W}}(s) - r_{\mathcal{W}}^\pi(s) \leq \varepsilon_1$ for all $s$. Therefore, $\hat{r}_{\mathcal{W}}(s) + \varepsilon_1 \geq r_{\mathcal{W}}^\pi(s)$ for all s and $(1 + \beta)\langle \rho^\pi, r_{\mathcal{W}}^\pi \rangle \leq (1 + \beta)\langle \rho^\pi, \hat{r}_{\mathcal{W}} + \varepsilon_1 \rangle$. So:

$$
(1 + \beta)\langle \rho^\pi, r_{\mathcal{W}}^\pi \rangle - (1 - \beta)\langle \mu_n, r_{\mathcal{W}}^\pi \rangle \leq (1 + \beta)\langle \rho^\pi, \hat{r}_{\mathcal{W}} + \varepsilon_1 \rangle - (1 - \beta)\langle \mu_n, r_{\mathcal{W}}^\pi \rangle
$$

$$
\leq (1 + \beta)\langle \rho^\pi, \hat{r}_{\mathcal{W}} \rangle - (1 - \beta)\langle \mu_n, r_{\mathcal{W}}^\pi \rangle + \varepsilon_1(1 + \beta)
$$

By definition $\langle \rho^{\pi'}, \hat{r}_{\mathcal{W}} \rangle - \langle \rho^\pi, \hat{r}_{\mathcal{W}} \rangle = \varepsilon_2$. So:

$$
(1+\beta)\langle \rho^\pi, \hat{r}_{\mathcal{W}} \rangle - (1-\beta)\langle \mu_n, r_{\mathcal{W}}^\pi \rangle + \varepsilon_1(1+\beta) = (1+\beta)\langle \rho^{\pi'}, \hat{r}_{\mathcal{W}} \rangle - (1-\beta)\langle \mu_n, r_{\mathcal{W}}^\pi \rangle + \varepsilon_1(1+\beta) - \varepsilon_2
$$

As previously, since $\|\hat{r}_{\mathcal{W}} - r_{\mathcal{W}}^\pi\|_\infty \leq \varepsilon_1$, one has $|\hat{r}_{\mathcal{W}}(s) - r_{\mathcal{W}}^\pi(s)| \leq \varepsilon_1$ for all $s$. And hence $\hat{r}_{\mathcal{W}}(s) \leq r_{\mathcal{W}}^\pi(s) + \varepsilon_1$. Consequently:

$$
(1 + \beta)\langle \rho^{\pi'}, \hat{r}_{\mathcal{W}} \rangle - (1 - \beta)\langle \mu_n, r_{\mathcal{W}}^\pi \rangle + \varepsilon_1(1 + \beta) - \varepsilon_2 \leq (1 + \beta)\langle \rho^{\pi'}, r_{\mathcal{W}}^\pi + \varepsilon_1 \rangle - (1 - \beta)\langle \mu_n, r_{\mathcal{W}}^\pi \rangle + \varepsilon_1(1 + \beta) - \varepsilon_2
$$

$$
\leq (1 + \beta)\langle \rho^{\pi'}, r_{\mathcal{W}}^\pi \rangle - (1 - \beta)\langle \mu_n, r_{\mathcal{W}}^\pi \rangle + 2\varepsilon_1(1 + \beta) - \varepsilon_2
$$

Here, $(1+\beta)\langle \rho^{\pi'}, r_{\mathcal{W}}^\pi \rangle - (1-\beta)\langle \mu_n, r_{\mathcal{W}}^\pi \rangle$ is not exactly equal to $\mathcal{W}(\rho^{\pi'}, \beta\rho^{\pi'} + \mu_n(\beta-1))$ because $r_{\mathcal{W}}^\pi$ was in fact a solution to the problem $\mathcal{W}(\rho^\pi, \beta\rho^\pi + \mu_n(\beta - 1))$. Hopefully, since by definition, $r_{\mathcal{W}}^\pi$ is 1-Lipshitz continuous with respect to the temporal distance, we have necessarily:

$$
(1 + \beta)\langle \rho^{\pi'}, r_{\mathcal{W}}^\pi \rangle - (1 - \beta)\langle \mu_n, r_{\mathcal{W}}^\pi \rangle \leq \max_{\substack{f \\ \text{s.t.} \\ \|f\| \leq 1}} (1 + \beta)\langle \rho^{\pi'}, f \rangle - (1 - \beta)\langle \mu_n, f \rangle
$$

$$
\leq \mathcal{W}(\rho^{\pi'}, \beta\rho^{\pi'} + \mu_n(\beta - 1))
$$

Putting it all together we have:

$$
\mathcal{W}(\rho^\pi, \beta\rho^\pi + \mu_n(\beta - 1)) \leq \mathcal{W}(\rho^{\pi'}, \beta\rho^{\pi'} + \mu_n(\beta - 1)) + 2\varepsilon_1(1 + \beta) - \varepsilon_2
$$

Finally, if $\varepsilon_2 - 2\varepsilon_1(1 + \beta) \geq 0$ we guarantee that $\mathcal{W}(\rho^{\pi'}, \beta\rho^{\pi'} + \mu_n(\beta - 1))$ is larger than $\mathcal{W}(\rho^\pi, \beta\rho^\pi + \mu_n(\beta - 1))$, which concludes the proof. $\qquad\square$

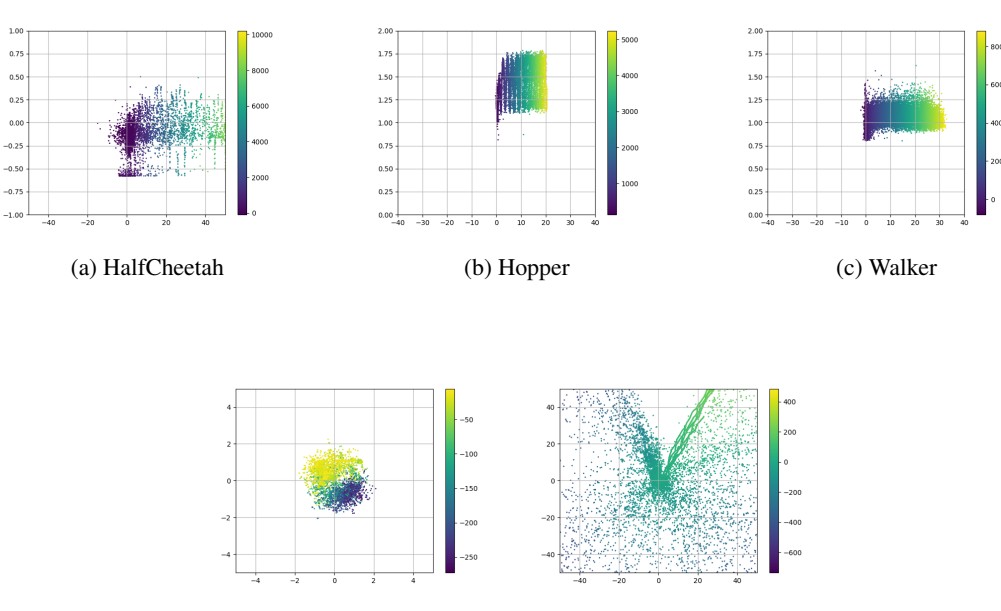

Figure 4: Euclidean coordinates of the states contained in the buffers used by RAMP with the color indicating the output of $f_\phi$.

## E    DENSITY MEASURES

Figure 4 illustrates the reward model employed by RAMP$_\mathcal{W}$ within the locomotion tasks.

## F    CONSTRUCTING $\mathcal{D}_{\mu_n}$

One approach to building $\mathcal{D}_{\mu_n}$ is to initialize a buffer $\mathcal{D}_{\mu_0}$ with a fixed size using a random policy and then replace elements in that list to incorporate the agent experiences. Specifically, each time the policy is updated, the sample $\mathcal{D}_{\mu_n}$ regrouping the past experiences of the agent should be updated. For some algorithms, such as PPO (Schulman et al., 2017), updating $\mathcal{D}_{\mu_n}$ is straightforward because the distribution $\rho^\pi$ can be clearly identified between updates. In this case, $\mathcal{D}_{\mu_{n+1}}$ is constructed by combining a proportion $\beta$ of samples from $\mathcal{D}_{\rho^\pi}$ with a proportion $(1-\beta)$ of samples from $\mathcal{D}_{\mu_n}$ each time the policy is updated. For SAC, where the policy is updated every step in the environment, each new state serves as a single sample for that policy. To determine whether a state should be added in $\mathcal{D}_{\mu_{n+1}}$, an accept-reject procedure using a Bernoulli distribution with parameter $\beta$ is applied for each step. The current state replaces a random element in the list $\mathcal{D}_{\mu_n}$ when accepted.

## G    REMINDER ON CONTRASTIVE LEARNING

By definition, a contrastive learning problem is a specific type of classification problem. This scenario usually involves a neural network $f_\phi : \mathcal{S} \to \mathbb{R}$ that employs a sigmoid activation function $\sigma$ in its final layer, focusing on two classes. Given a distribution $S$ over $\mathcal{S}$, a standard classification problem can be expressed as:

$$\mathcal{L}(\phi) = \underset{s \sim S}{\mathbb{E}}[l(y(s), \sigma(f_\phi(s)))]$$

where $y(s)$ assigns a label of 0 or 1 to any sample $s$, and $l$ represents a loss function which attains 0 when $y(s) = \sigma(f_\phi(s))$. Typically, $l$ is a Cross-Entropy loss, which can be retrieved via a Kullback-Leibler divergence between two Bernoulli distributions $\mathcal{B}$, parameterized respectively by $y(s)$ and

$\sigma(f_\phi(s))$:

$$\mathcal{L}(\phi) = \mathop{\mathbb{E}}_{s \sim S} \left[ \text{KL}(\mathcal{B}(y(s)) || \mathcal{B}(\sigma(f_\phi(s)))) \right]$$

$$= \mathop{\mathbb{E}}_{s \sim S} \left[ y(s) \log \left( \frac{y(s)}{\sigma(f_\phi(s))} \right) + (1 - y(s)) \log \left( \frac{1 - y(s)}{1 - \sigma(f_\phi(s))} \right) \right]$$

$$= \mathop{\mathbb{E}}_{s \sim S} \left[ \underbrace{\text{H}(\mathcal{B}(y(s)) | \mathcal{B}(\sigma(f_\phi(s))))}_{\text{Cross-entropy}} - \underbrace{H(\mathcal{B}(y(s)))}_{\text{Shannon entropy}} \right]$$

$$= \mathop{\mathbb{E}}_{s \sim S} \left[ \underbrace{\text{H}(\mathcal{B}(y(s)) | \mathcal{B}(\sigma(f_\phi(s))))}_{\text{Cross-entropy}} - \underbrace{H(\mathcal{B}(y(s)))}_{\text{cte}} \right]$$

In contrastive learning, the classification problem typically involves two random variables, $S_+$ and $S_-$, and the target function $y(s)$ defined as:

$$y(s) = \begin{cases} 1 \text{ if } s \sim S_+ \\ 0 \text{ if } s \sim S_- \end{cases}$$

When the probabilities of sampling data from $S_+$ and $S_-$ are equal, the classification loss can be reformulated as:

$$\mathcal{L}(\phi) = - \mathop{\mathbb{E}}_{s \sim S_+} \left[ \log \left( \sigma(f_\phi(s)) \right) \right] - \mathop{\mathbb{E}}_{s \sim S_-} \left[ \log \left( 1 - \sigma(f_\phi(s)) \right) \right]$$

## H   WHY IS MAXIMIZING $\text{D}_{\text{KL}}(\rho^\pi || \beta\rho^\pi + (1 - \beta)\mu_n)$ REALLY RUNNING AWAY FROM THE PAST?

This study posits that maximizing the following Kullback-Leibler divergence:

$$\text{D}_{\text{KL}}(\rho^\pi || \beta\rho^\pi + (1 - \beta)\mu_n)$$

with respect to $\pi$, generates a distribution $\rho^\pi$ that deviates maximally from the current mixture of past distributions $\mu_n$. However, the distribution used for comparison, $\beta\rho^\pi + (1-\beta)\mu_n$, also includes $\rho^\pi$ itself. While one might consider maximizing $\text{D}_{\text{KL}}(\rho^\pi || \mu_n)$, this alternative KL divergence is not always well-defined, particularly when the supports of $\rho^\pi$ and $\mu_n$ are disjoint. Notably, Proposition 4 indicates that the distributions solving the problem $\arg \max_{\rho} \text{D}_{\text{KL}}(\rho || (\beta\rho + (1-\beta)\mu_n))$ are contained within the solutions to the problem $\arg \max_{\rho} \text{D}_{\text{KL}}(\rho || \mu_n)$.

**Proposition 4.** *Let $\mu_n \in \Delta(S)$ such that there exists $s \in \mathcal{S}$ for which $\mu_n(s) = 0$. Let $\Delta_1^*(S) = \arg \max_{\rho} D_{KL}(\rho || (\beta\rho + (1 - \beta)\mu_n)$ and $\Delta_2^*(S) = \arg \max_{\rho} D_{KL}(\rho || \mu_n)$. The set of distributions $\Delta_1^*(S)$ is a subset of the distributions $\Delta_2^*(S)$ i.e. $\Delta_1^*(S) \subset \Delta_2^*(S)$*

*Proof.* Let's define $\Delta_1^*(S) = \arg \max_{\rho} \text{D}_{\text{KL}}(\rho || (\beta\rho + (1 - \beta)\mu_n)$, $\Delta_2^*(S) = \arg \max_{\rho} \text{D}_{\text{KL}}(\rho || \mu_n)$ and $\Delta_3^*(S) = \{\rho \in \Delta(\mathcal{S}) | \forall s \in \mathcal{S}, \rho(s) > 0 \implies \mu_n(s) = 0\}$.

The first step is to show $\Delta_1^*(S) = \Delta_3^*(S)$. By definition:

$$\text{D}_{\text{KL}}(\rho || (\beta\rho + (1 - \beta)\mu_n) = \int_\mathcal{S} \rho(s) \log \left( \frac{\rho(s)}{\beta\rho(s) + (1 - \beta)\mu_n(s)} \right) \text{d}s$$

Since $\mu_n(s) \geq 0 \, \forall s \in \mathcal{S}$:

$$\log \left( \frac{\rho(s)}{\beta\rho(s) + (1 - \beta)\mu_n(s)} \right) \leq \log \left( \frac{1}{\beta} \right)$$

And this upper bound is reached whenever $\mu_n(s) = 0$. Additionally, as $\beta \in (0,1)$, $\log\left(\frac{1}{\beta}\right) \geq 0$. So maximizing this $D_{KL}$ corresponds to having distributions $\rho$ that have a support as disjoint as possible from the support of $\mu_n$ (because $\log\left(\frac{1}{\beta}\right)$ would be integrated over greater portion of $\mathcal{S}$). Therefore, the set of optimal solutions $\Delta_2^*(S)$ may be rewritten as $\{\rho \in \Delta(\mathcal{S}) | \forall s \in \mathcal{S}, \rho(s) > 0 \implies \mu_n(s) = 0\}$ which is by definition $\Delta_3^*$.

Note: The definition of $\rho$ depends on an initial distribution $\delta_0$. Since the successive $\rho_{k \in [0,n]}$ are used to fill $\mu_n$, the supports of $\mu_n$ and $\rho$ is probably not disjoint and $\{\rho \in \Delta(\mathcal{S}) | \forall s \in \mathcal{S}, \rho(s) > 0 \implies \mu_n(s) = 0\}$ is empty.

Additionally, since $\forall \rho \in \Delta_3^*(S)$ $D_{KL}(\rho||\mu_n) = +\infty$ (because $\log(\frac{c}{0}) = +\infty$), and every element of $\Delta_3^*$ is in $\Delta_2^*$ and so $\Delta_1^*(S) \subset \Delta_2^*(S)$ which ends the proof. $\square$

# I  BENCHMARKS

**Mazes**

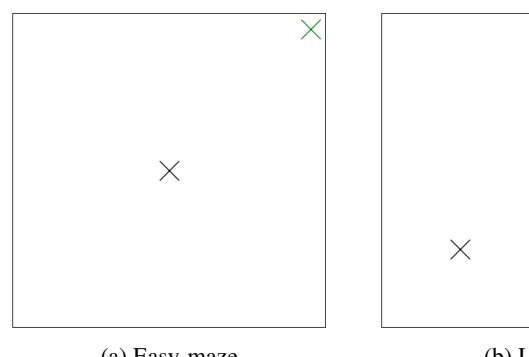 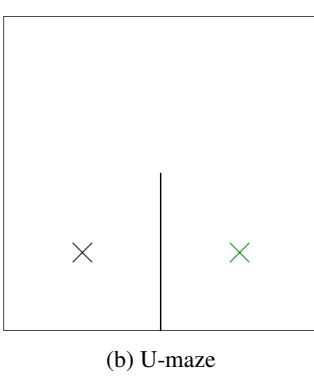 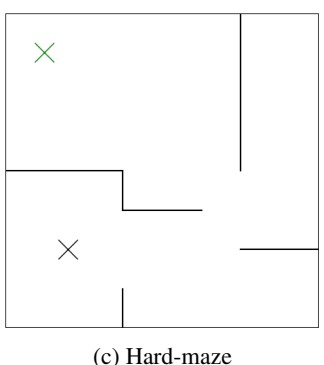

|  (a) Easy-maze  |  (b) U-maze  |  (c) Hard-maze  |

Figure 7 illustrates all the benchmarks used in the experiments, including our custom set of mazes. The dynamics of the mazes are based on a simple Euler integration: $s_{t+1} = s_t + a_t \, dt$, with $dt = 0.001$, $(s_t, a_t) \in [-1,1]^2$, and the initial state $s_0$ located at the black cross while the goal state $s_g$ is located at the green cross. The reward model used is $r(s, a, s') = \|s_g - s\|_2 - \|s_g - s'\|_2$.

**Locomotion Tasks**

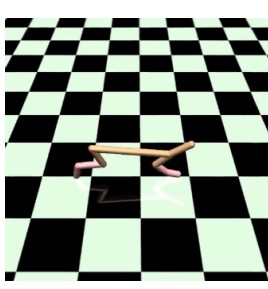
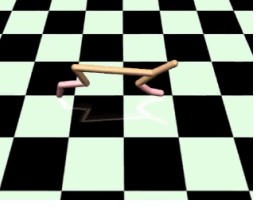
(a) HalfCheetah

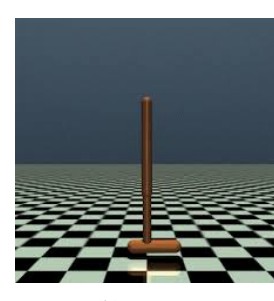
(b) Hopper

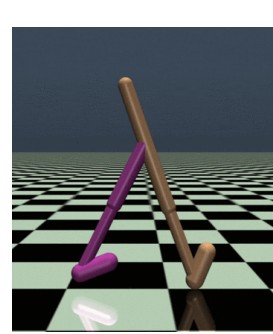
(c) Walker

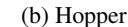

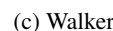

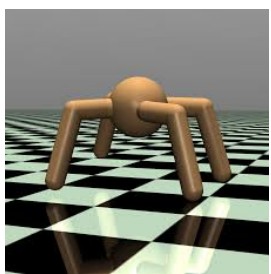
(d) Humanoid

(e) Ant

**Robotic Tasks**

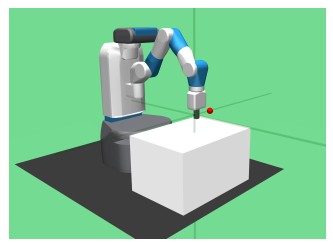
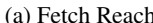
(a) Fetch Reach

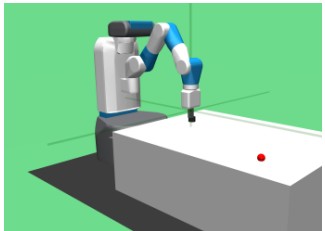
(b) Fetch Slide

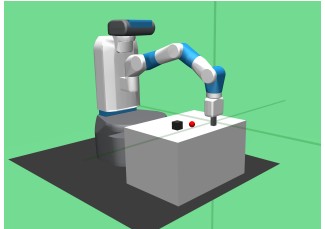
(c) Fetch Push

Figure 7: Set of tasks.

