# OpenReview forum: "Exploration by Running Away from the Past"
_ICLR.cc/2025/Conference — Submitted to ICLR 2025_

### Official Review · Reviewer_nQGL · 2024-11-01

**Soundness:** 3
**Presentation:** 3
**Contribution:** 2
**Rating:** 3
**Confidence:** 4

**Summary:**

The paper proposes a new intrinsic exploration objective for maximizing state entropy. The objective uses a discounted mixture of past state occupancy measures and encourages policies that maximize distance from the discounted mixture. As statistical distance, the KL divergence and Wasserstein distance are used. The experiments are evaluated on state-based RL environments, where state coverage and episodic returns are used to demonstrate the performance gain of the proposed approach.

**Strengths:**

The paper is written well, and the proposed intrinsic exploration objective is novel. The use of Wasserstein distance instead of the typical KL divergence is an interesting/novel choice.

**Weaknesses:**

1) Prior Work/Baselines: The paper misses several crucial works on intrinsic exploration (c.f., 1, 2, 3 for a survey). Particularly, there are works that use the model epistemic uncertainty/disagreement as an intrinsic reward which works well in practice and also scales favorably (4., 5., 6.).
2) Theory: In particular, the model epistemic uncertainty is theoretically a well-studied objective (7., 8.). In 8, the authors derive a connection between maximizing the model epistemic uncertainty and maximizing information gain/conditional entropy of the trajectories, while also showing convergence for sufficiently smooth dynamics.
3) Unclear motivation: Given the theoretical and experimental strengths of the method discussed above, its unclear to me what particular gap the authors are trying to address with their intrinsic reward. I'd appreciate the authors elaborating further on this. Furthermore, I think all the aforementioned works should be discussed in the paper and in particular one of the baselines should use the model epistemic uncertainty as the intrinsic reward. Perhaps one weakness the authors might raise is that the aforementioned works are computationally more expensive as they have to learn an ensemble of networks to quantify disagreement. However, this should also be empirically shown in the experiments (as the proposed method also learns a model to estimate the intrinsic reward).
4) Hyperparameters are not provided in the paper, which makes it difficult for me to assess how sensitive the results are to the choice of hyperparams. In particular, I am curious about how $\beta$ affects the performance of the algorithm. How can we appropriately select $\beta$? Furthermore, doesn't the method suffer from sample inefficiency for large values for $\beta$, i.e., when lots of data from the buffer is discarded?
5) Scalability: Its unclear to me whether the proposed method would scale reasonably well to more high-dimensional settings such as POMDPs/visual-control tasks (note that 5, 6 also work for POMDPs). Could the authors elaborate further on this?

I am happy to raise my score if my concerns above are addressed.

1. https://arxiv.org/abs/2109.00157
2. https://www.sciencedirect.com/science/article/pii/S1566253522000288?casa_token=ScYOIGv6D2wAAAAA:buNFoXMZLqPiWzo0CLpe3K-ac_nxundN5855FT0QwSnE6jhpm6VwPFS0UHyt1E9WXJePruqZsg
3. https://www.mdpi.com/1099-4300/25/2/327
4. https://arxiv.org/pdf/1906.04161
5. https://arxiv.org/abs/2005.05960
6. https://arxiv.org/abs/2110.09514
7. https://arxiv.org/pdf/2006.10277
8. https://arxiv.org/pdf/2306.12371

**Questions:**

See Weaknesses.

---

> ### Author Response · Authors · 2024-11-21
>
> # Answer to Reviewer nQGL
> We thank the reviewer for their thorough feedback.
>
> ## Related Work on Epistemic Uncertainty
> We appreciate Reviewer nQGL's suggestion to discuss the relationship between our work and methods that leverage epistemic uncertainty. We agree that this is an interesting aspect of exploration, and we plan on including a discussion of these methods in the related work section of the revised manuscript. However, the experimental comparison uses 10 baseline methods that we believe represents a wide range of the exploration literature, and adding further experimental comparisons is out of the scope of this article.
> ## Hyperparameters
> As discussed in the response to reviewer YHsc, we plan on providing a more comprehensive explanation of the hyperparameters used in the appendix and will discuss the impact of hyperparameter choices.
> ## Exploration in POMDPs
> Our primary motivation for reframing entropy maximization through the KL divergence was to use a density estimator known to scale effectively in high-dimensional problems, such as visual control tasks. More specifically, by framing the problem as KL maximization, the density model that makes the most sense to use is a classifier, which is known to scale well in high-dimensional settings. Although we do not test our method in POMDPs, we thank the reviewer for the suggestion and will include a discussion on POMDPs as a prospect for future work.

---

> > ### Comment · Reviewer_nQGL · 2024-11-21
> > **Response to the author's rebuttal**
> >
> > I thank the authors for their response.
> >
> > **Epistemic Uncertainty**: While I appreciate the authors including epistemic uncertainty-based exploration methods in the discussion. I do not understand how they are considered out of scope for this paper. Particularly, given the theoretical connection between maximizing information gain and entropy and that the epistemic uncertainty-based methods are widely applied for intrinsic exploration. Moreover, I think these methods constitute a strong baseline that should be compared against.
> >
> > **Hyperparameters**: Thanks for including this. I would like to look at the proposed changes before reevaluating my score.
> >
> > **Exploration in POMDPs**: While I can follow the arguments made by the reviewer, I would require empirical evaluation to be fully convinced. However, I acknowledge that this might be out of the scope of this work.
> >
> >
> > Overall, I would like to keep my score as I believe my concerns, particularly for the first two points above, are not yet fully addressed.

---

### Official Review · Reviewer_YHsc · 2024-11-03

**Soundness:** 2
**Presentation:** 3
**Contribution:** 2
**Rating:** 5
**Confidence:** 4

**Summary:**

The paper proposes an exploration paradigm of "running away from the past" (RAMP), which encourages the RL algorithm to generate trajectories in distribution different from the past. This is instantiated as an intrinsic exploration bonus that estimates the discrepancy between the current and past visitation density. They show improvements on a few benchmark deep RL algorithm, showcasing the potential for this approach.

**Strengths:**

The strength of the paper lies in a fairly clear presentation of the motivation and methodology. The idea of "running away from the past" is not strictly novel but the paper proposes an algorithmically viable way to instantiate such an idea. The paper presents a fairly clear math formulation and has carried out ablations on choices of the algorithmic designs. The experimental ablation also seems fairly comprehensive.

**Weaknesses:**

The idea of "running away from the past" is not strictly novel. From a theoretical standpoint, running away from old trajectories might not always be optimal and it is not clear theoretically what is gained by adopting such an approach. From an empirical standpoint, the ablations are carried out on the continuous control tasks, most of which do not seem to require extensive exploration to solve. It is not very clear if the claimed gains are really due to the exploration bonus, or some other unknown side effect.

**Questions:**

### === Theoretical gains of running away from the past ===

I think the idea of RAMP makes sense in that for the algorithm to explore, it must do something different from the past. However there is always a trade-off in practice and one must balance exploration vs. exploitation, a factor that is heavily environment dependent. There are simply environments where such exploration is not needed at all, while others where exploration is needed.

At this point, deep RL literature has already accumulated a large varieties of exploration methods, each dedicated to a specific domain. I think it will be valuable if a more general purpose method such as RAMP can characterize the theoretical gains achieved by just maximizing the distributional divergence between the current policy and previous data distributions.

### === Intrinsic reward alone ===

Table 1 shows the max performance that can be achieved by different exploration methods using just the intrinsic reward. In a sense, it measures how extreme the performance can reach by just optimizing for the exploration bonus. It is quite a surprise to me that RAMP's intrinsic reward leads to max gain very much higher than most methods. I think it might also be beneficial to plot the distribution of rewards achieved by different methods, to robustly measure the range of performance achievable. After all, max is not a very robust estimate of the possible performance obtained by the policy.

It also seems that Wasserstein based approach is much higher than KL - given that both are motivated by the RAMP narrative, it seems that the specific choice of metric is also very critical to the algorithmic performance. Do you think the underlying metric that defines Wasserstein distance is also critical, ie L2 vs L1 distance. Such ablations will be quite valuable to practitioners.

### === Extrinsic reward ===

In Table 2 where extrinsic rewards are combined, it seems that KL RAMP is better than Wasserstein RAMP in general, which is in opposite to the results in Table 1 where Wasserstein RAMP is generally better. Can you elaborate more on this?

Also in general in continuous control tasks, it seems that exploration is not a defining factor to the final performance - as opposed to certain exploration heavy tasks in atari suites. As a result, it is not very clear if the gains in performance are due to the exploration bonus itself or rather due to some other confounding factors as a result of adding the corresponding loss.

In practice, how would you choose the exploration vs. exploitation trade-off factor ($\lambda$ and $\beta$ factors in the algorithm), and are the algorithmic performance sensitive to the choice of such hyper-parameters?

---

> ### Author Response · Authors · 2024-11-21
>
> # Answer to Reviewer YHsc
> We thank the reviewer for their thorough feedback. To ensure we fully understand each of the comments, we would appreciate it if the reviewer could confirm that our interpretations are correct.
> ## Intrinsic Reward Alone
> * **Table 1**:
> Table 1 shows the final coverage reached by each method on each environment across 5 different seeds.
> * **Distance Used for the Wasserstein Distance**:
> The distance used for the Wasserstein distance is the temporal distance, as explained in the paper. The temporal distance (while not strictly adhering to the properties of a distance function) is sensible in any environment that can be represented as a Markov Chain, which makes the intuition behind this algorithm generalizable to a wide range of environments. As the goal of this study is to derive a general algorithm applicable to a broad range of environments, we believe that the temporal distance is a suitable choice for the general case. Further study on the impact of using $l_2$, $l_1$, and temporal distances for the Wasserstein distance could benefit specific applications, but we find it out of the scope of this article.
> ## Hyperparameters
> We agree that further details on hyperparameters would be useful. We purposefully chose a small set of hyperparameters to explore to underlie the robustness of the model to different hyperparameter choices. A more exhaustive study of the possible hyperparameter space would benefit the paper and we will consider it, depending on computational costs. We will, however, provide a more detailed explanation of the hyperparameters used in the appendix.

---

> > ### Comment · Reviewer_YHsc · 2024-11-27
> > **reply**
> >
> > Thank you to the authors for the reply.
> >
> > Thanks for clarifying on some technical details that I missed from first reading the paper. It will indeed be helpful having access to a more detailed explanations of the hyper-parameters used for experiments.
> >
> > I will adjust my scores after the discussion phase.

---

### Official Review · Reviewer_FV3w · 2024-11-03

**Soundness:** 1
**Presentation:** 2
**Contribution:** 3
**Rating:** 3
**Confidence:** 3

**Summary:**

The authors present a new algorithm for learning policies where the marginal distribution of states in a trajectory of length $T$ has a high entropy. Their method consists in iteratively maximizing intrinsic reward bonuses that measure a distance (metric) between the distribution of states of the current policy, and a geometric weighting of the distributions of states for the previous policies. That objective finds a motivation from an information theory property. Experiments compare the use of the KL-divergence and the Wasserstein distance to other algorithms.

**Strengths:**

1. The problem addressed is important to the community.
2. The new objective function is theoretically motivated and provides new insights to compute good exploration policies.

**Weaknesses:**

1. In Section 2, different justifications for introducing the learning objective pursued by the agent are wrong or weak in several aspects:

a. The justification line 108 for going from equation (1) to equation (2) is in my opinion wrong. Using the entropy of the policy as proxy to the entropy of the state distribution is a huge approximation. Maximizing the entropy of the policy does not provide a good state coverage in general nor in most practical cases. Note that if it was sufficient to maximize the entropy of the policy to get a uniform distribution of states, it would not be necessary to introduce a complex algorithm as the authors do.

b. Line 128 authors justify to use the Wasserstein distance instead of the KL-divergence as the KL does not account for a potential geometry in the state space. This fact result from the original choice to define as exploration objective the entropy over the state space, which does not account for a potential geometry of the state space. So by choosing to maximize the Wasserstein distance instead of the KL, the authors change the original hypothesis that that the objective is to have high state entropy. While it can be discussed that it is a potential better framework to account for some geometry, it makes most of the previous mathematical justifications irrelevant.

2. The authors claim in Section 3.4 that it is sufficient to optimize with any RL algorithm the reward model from Section 3.2 or Section 3.3 to maximize the objective equation (2) or equation (4). It is equivalent to neglecting the entropy of the policy. Authors, nevertheless, eventually use SAC, which is an algorithm that regularizes the MDP rewards with the log-likelihood of actions. This should be clarified.

3. Only the final values are reported in the experimental section. From my personal experience, complex exploration methods may be unstable, and the learning curves provide important insights. Adding them in the paper would make the results more trustworthy.

4. In the experiments, there is no statistical evidence that the method at hand outperforms the concurrent methods. Most confidence intervals overlap.

5. I think that the related work should include [1, 2], and probably other, more recent, works.

**Questions:**

1. Line 81, authors state that the methodology 'seamlessly' generalizes when T tends to infinity. From a theoretical perspective, does the limit exist without additional assumptions on the markov chain created from the MDP and the policy? From a practical point of view, are there any limitation to apply the method when T is large?
2. Is a policy with maximum state entropy an optimal solution to the objective function that is maximized?
3. There is a typo in equation (2): $\rho^\pi$ should be $\rho^\pi(s)$.
4. The optimization problems for computing the intrinsic reward functions seem to be on-policy, is it the case? If it is the case, does it eventually result to on-policy optimization of control policies? If it is the case, it is worth mentioning that when used in combination with an off-policy RL algorithm for maximizing the intrinsic reward, addition interactions with the MDP are required making the modified algorithm on-policy. This point should be clear in Section 3.4.
5. Could the authors clarify the arguments from paragraph line 329? I understand the philosophy of maximizing a lower bound on the entropy instead of directly maximizing the entropy. Yet, I think that both approaches incrementally improve the Shannon entropy, in opposition to the first sentence of the paragraph. I don't understand the argument of the generalization across behaviours.

---

> ### Author Response · Authors · 2024-11-21
>
> # Answer to Reviewer FV3w
> We thank the reviewer for their thorough feedback.
> ## Section 2
> ### Assumption on Entropy of the Occupancy Measure
> We agree with Reviewer FV3w regarding the assumption that maximizing the expected entropy of the policies on the occupancy measure leads to high entropy of the occupancy measure. However, we explicitly state in the paper that this "hypothesis may not universally apply across all environments". Overall, our assumption is that the Kullback-Leibler divergence term in the lower bound is the most significant, and thus maximizing the entropy of the occupancy measure through the proposed proxy is not the primary driver of improvement. We will clarify this further in the next version.
>
>
> ### From KL to the Wasserstein Distance
>
> The paper initially aims to maximize Shannon entropy, which leads to maximizing the KL divergence between the current occupancy measure $\rho^{\pi}$ and the mixture $\beta\rho^{\pi} + (1-\beta)\mu_n$, where $\mu_n$ represents the past occupancy measures. It is only upon introducing this KL divergence that we consider the potential advantages of using a divergence that captures the overall geometry of the state space. Could the reviewer clarify why this would make the derivation of the KL divergence objective irrelevant?
> Given the objective involving a divergence measure between two distributions, we believe that the study of alternative methods to quantify the gap between distributions is founded, but would enjoy further clarification.
> ## Using the SAC Algorithm
> Indeed, we used SAC to maximize all reward models. We agree that the regularization used in SAC aligns well with our objective of maximizing the expected entropy of the policies on the occupancy measure, and we will clarify this in the paper.
> ## Best Policy
> When using intrinsic reward models to maximize an extrinsic reward, the learning curve can oscillate as the agent explores new areas, resulting in high intrinsic reward values. As a result, we believe that the most relevant way to benchmark these algorithms is to consider the best policy discovered since the beginning of training. However, we agree that analyzing the temporal evolution of the learning curves could provide additional insights into the exploration behavior of the algorithms.
> ## Statistical Significance: Exploration and Exploitation Assessment of the Results
> We agree with the reviewer that the statistical significance of the results should be clearly presented. There is a statistically significant difference between the methods in many environments and tasks, although this may not be sufficiently apparent in the current version. We propose to highlight in bold the method for which the paired t-test with the second-best method indicates a significant difference.
> ## Related Work
> The reviewer cites [1] and [2] but it appears that the references were not included. Would it be possible to include them?

---

> > ### Comment · Reviewer_FV3w · 2024-11-26
> > **Follow-up**
> >
> > Thank you for responding to my review. I will clarify some elements.
> >
> >
> > 1. I don't believe that the KL divergence objective is irrelevant at all, sorry for the misinterpretation. I believe that the argument about the 'geometry ' is not completely correct. In my opinion, the KL objective is one you use when you do not make any assumption on the geometry. When you assume there exist a structure, you may then rely on another objective. But I don't think that when you assume there is no structure, it makes sense to use the Wasserstein objective, does it?
> > 2. Concerning the additional work, I was referring to [1] and [2], but many more can be found.
> >
> > Overall, I don't think you response fully answer my remarks and I believe that yout manuscript would benefit from deeper modifications.
> >
> > [1] Zhaohan Daniel Guo, Mohammad Gheshlaghi Azar, Alaa Saade, Shantanu Thakoor, Bilal Piot,
> > Bernardo Avila Pires, Michal Valko, Thomas Mesnard, Tor Lattimore, and R´emi Munos. Geometric
> > entropic exploration.
> >
> > [2] Chuheng Zhang, Yuanying Cai, Longbo Huang, and Jian Li. Exploration by maximizing r´enyi
> > entropy for reward-free rl framework.

---

### Official Review · Reviewer_65e4 · 2024-11-04

**Soundness:** 2
**Presentation:** 3
**Contribution:** 2
**Rating:** 3
**Confidence:** 4

**Summary:**

The paper proposes RAMP (Running away from the past), an RL-based method for performing state space exploration by approximately maximizing either the KL divergence or Wasserstein distance between the current policy's state occupancy measure and the discounted sum of the state occupancy measures of all previous policies. This scheme aims to ensure that the state space coverage provided by the next policy is always maximally different from that provided by previously policies. The paper develops the RAMP method by deriving tractable proxies for these divergences, proposing reward models for each that can be used in conjunction with an RL algorithm, providing related approximation bounds, providing estimation schemes for each reward model based on existing work, and finally combining these steps to propose RAMP. Experimental results are provided that quantitatively illustrate what the reward models look like, compare RAMP with other intrinsic exploration approaches using a certain notion of state space coverage on a variety of tasks, and indicate that RAMP can be used as an exploration aid to accelerate extrinsic reward learning tasks.

**Strengths:**

Though the problem of state space exploration is very extensively covered in the RL literature, the proposed RAMP method provides what appears to be a novel approach to accelerating state space coverage. Due to its strategy of choosing policies maximizing divergence of state space coverage from that achieved by previous policies, it makes sense that RAMP will be more effective at rapidly exploring the state space than existing unsupervised RL methods (e.g., APT, SMM, Proto-RL) that simply maximize state occupancy measure entropy, and the experiments provide some support to this. Moreover, though the actual learning procedure used in RAMP is essentially a combination of existing techniques ([Eysenbach et al., 2020] for $r_{KL}$, [Durugkar et al., 2021] for $r_{W}$, and SAC [Haarnoja et al., 2018]), the combined approach detailed in Sec. 3.4 and Algorithm 1 appears to be novel and is interesting, and the fact that both KL-divergence and Wasserstein distance versions of RAMP are provided adds to its flexibility and significance. For these reasons, RAMP is likely of interest to the community and definitely merits further investigation.

**Weaknesses:**

Despite the strengths discussed above, I have concerns about the experimental evaluation and theoretical results:
1. Most importantly, the "state coverage" performance metric upon which the comparisons of Sections 5.2 and A.1 rely is insufficiently justified as a good proxy for measuring exploration and for making fair comparisons between the algorithms considered. As described in the third paragraph of Sec. 5.2, this metric is obtained by discretizing the space of Euclidean (x-y or x-y-z) coordinates of the agent's state space, recording whether each grid cell has been visited or not during training, then returning the percentage of the grid cells that have been visited. There are two main issues with using this notion of state coverage as a proxy for exploration. First, the state space dimensions in most of the environments are far larger than 2 or 3 (e.g., 18 for HalfCheetah, 113 for Ant), and, for many of these environments, pose information other than location in Euclidean space (e.g., joint angles, velocities) is far more important for learning to operate within the environment and for specific downstream tasks. Second, recording only whether a grid cell has been visited or not ignores more complex visitation behavior, such as the empirical state visitation frequency defined at the beginning of Sec. 2. To render the state coverage metric used more meaningful, it would be helpful to include ablations over the other dimensions of $S$ or comparison with other coverage notions, such as Shannon entropy of the empirical state visitation frequency.
2. Implementation details for the RAMP algorithm, the algorithms compared with, the discretization used in the state coverage metric, and other aspects of the experiments are not provided. The experimental results are therefore not reproducible in their current form. In addition, across all experiments, the lack of implementation details makes it difficult to assess the fairness of comparison with existing methods and even the comparisons between $RAMP_{KL}$ and $RAMP_{W}$. This makes it difficult to evaluate the significance of the experimental results, weakening the overall contribution. To remedy these issues, a thorough description of the implementation details is needed.
3. The qualitative results in Sec. 5.1 are difficult to understand, leaving the practical differences between $r_{KL}$ and $r_W$ unclear. See the questions below for specific concerns.
4. The connection between Theorems 2 and 3 and the rest of the paper is unclear, and the assumptions made are so strong as to immediately imply the results. For the former concern, a description of what $\pi$ and $\pi'$ of Theorems 2 and 3 correspond to in the RAMP method is missing, making it unclear how the results are meant to be applied. Regarding the second concern, it is assumed variously that $|| \rho^{\pi} - \rho^{\pi'} || \leq \varepsilon_0$, $|| \hat{r} - r^{\pi} || \leq \varepsilon_1$, and that the average reward $J_{\hat{r}}(\pi') = \langle \rho^{\pi'}, \hat{r} \rangle$ is sufficiently larger than $J_{\hat{r}}(\pi) = \langle \rho^{\pi}, \hat{r} \rangle$ to ensure that the desired inequalities hold. Under these assumptions, the proofs follow with some straightforward manipulation of inequalities. To make the results more consequential, it would be helpful to clarify how they are meant to be applied in the context of the paper, then weaken the assumptions accordingly.

**Questions:**

1. Why is the notion of state coverage used in the experiments a good proxy for evaluating and comparison exploration? What is lost by ignoring the remaining dimensions in each of the environments considered?
2. What is the value of $n$ in Fig. 2? Can you provide additional context about the rewards pictured in Fig. 2?
3. Why is $r_W$ better than $r_{KL}$ in Fig. 2? This is mentioned in the paragraph starting line 377, but remains unclear.
4. What do the colors represent in Fig. 3?
5. Do the figures in Sec. 5.1 provide any insight into what is happening in the rest of the state space? Why not consider a visualization technique for visualizing high-dimensional data, such as $t$-SNE or PHATE plotting, instead of projecting onto x-y space?
6. What do $\pi$ and $\pi'$ of Theorems 2 and 3 correspond to in the RAMP method and the remainder of the paper?
7. How do Theorems 2 and 3 apply to the rest of the paper?

**Important additional comment:** It is stated at several points throughout the paper (line 047, lines 325-327, 330-332, 467-469, 682-684) that state entropy maximization methods like APT [Liu & Abbeel, 2021] rely on probability density estimation. This is not accurate: APT and similar methods (e.g., Proto-RL [Yarats et al., 2022]) leverage non-parametric $k$-nearest neighbor entropy estimators, allowing them to maximize (proxies of) state occupancy measure entropy while avoiding density estimation.

---

> ### Author Response · Authors · 2024-11-21
>
> # Answer to Reviewer 65e4
> We thank the reviewer for their thorough feedback.
> ## State coverage
> We acknowledge Reviewer 65e4's point that using Euclidean coordinates may not be the most accurate way to measure state space coverage. As the reviewer suggests, we quantify coverage by discretizing specific environment variables with a $10^2$ matrix using 10 steps for the $xy$ coordinates, and we count whether each cell is filled. However, as the reviewer noted, most environments have far more dimensions; for example, Ant has 111 dimensions. Even with only 10 cells per dimension, the number of cells needed to cover the state space is $10^{111}$, which is infeasible to compute. This practical limitation is why we use density estimators; otherwise, we would simply count the number of times a cell is visited.
> The reviewer also suggests estimating the Shannon entropy of past occupancy measures. While this could provide insights, it is also an imperfect measure of exploration. First, accurately estimating Shannon entropy would require discretizing all state space variables, which is infeasible. Second, even a precise entropy measure may not capture exploration perfectly, motivating our use of the Wasserstein distance.
> For example, the Shannon entropy of a uniform distribution is at its maximum, but this does not imply that the distribution's support fully covers the state space. That said, we agree that the temporal evolution of the Shannon entropy of the mixture of occupancy measures could be useful for identifying if the agent becomes stuck in a local minimum. We will include these results in the ablation study.
>
> ## Implementation details
> We agree with the reviewer that a thorough description of implementation details is essential. We propose to add two sections in the appendix: one detailing the hyperparameters and implementation specifics of our method, and another detailing those of the baselines. All implementations and hyperparameters are available in our code repository: [Link for the github repository](https://anonymous.4open.science/r/Exploration_by_running_away_from_the_past_039B).
>
> ## Qualitative results 5.1: Differences between $r_{\mathcal{W}}$ and $r_{\text{kl}}$
> The reviewer noted that Figure 2 does not clearly illustrate the difference between the Wasserstein distance and KL divergence. We propose to add a section in the appendix summarizing the definitions of the Wasserstein distance and KL divergence, along with their key differences.
> Our explanation of the two reward models may also be unclear, as we do not intend to suggest that one exploration method is superior to the other. The goal of Figure 2 is to show that maximizing $r_{\mathcal{W}}$ leads to a different exploration pattern of specific variables in the state space than maximizing $r_{\text{kl}}$. The intuition is that the density model of $r_{\mathcal{W}}$ is Lipschitz continuous with respect to temporal distance, meaning that in environments like HalfCheetah, the main variations of this reward model are triggered by major changes in Euclidean coordinates.
> Consequently, each algorithm induces distinct exploration behaviors that may be beneficial for different downstream tasks, as discussed in the paper. For instance, exploration driven by $r_{\mathcal{W}}$ may be more effective for goal-conditioned tasks where the objective is to reach a specific Euclidean position in the state space. In contrast, $r_{\text{kl}}$ may provide more fine-grained exploration, focusing on the exploration of the agent's different joint configurations, which can be useful in dense reward maximization tasks like the HalfCheetah running task (see Section 5.3).
>
> - **Figure 3 Interpretation**: In Figure 3, the colors represent the output of the density model estimating $r_{\mathcal{W}}$, as noted in the caption. We agree that adding a color bar would help clarify this and we will add it in the revised version.
> - **t-SNE Visualization**: Although we could use t-SNE to show which reward model emphasizes particular state space regions from a statistical perspective, our objective in this figure is to demonstrate that the exploration patterns induced by the two reward models differ, possibly impacting performance on different downstream tasks.

---

> > ### Author Response · Authors · 2024-11-21
> >
> > ## Proof in the paper
> > Theorems 2 and 3 were introduced to confirm that maximizing the reward models defined in the paper indeed maximizes the KL Divergence and Wasserstein distance described in Section 2. As a result, the assumptions underlying these theorems involve bounding estimation errors of the reward models, establishing a lower bound on the improvement step from policy $\pi$ to policy $\pi'$ (assuming the reward is maximized), and assuming limited change in the occupancy measure between policies $\pi$ and $\pi'$ (only for Theorem 2).
> > We agree with the reviewer that a clear discussion of the assumptions relative to the algorithm would be beneficial. We will add an appendix section to elaborate on these assumptions. Nonetheless, these theorems were primarily introduced to validate that the reward models proposed in the paper indeed maximize the KL Divergence and Wasserstein distance as defined in Section 2.
> > ## Important additional comment: on the use of density estimators
> > - **Probability density estimators**: Line 047: "these approaches often rely on probability density estimators, and finding a relevant density estimator for an environment can be challenging." Indeed, these approaches often rely on probability density estimators. APT does not rely on a probability density function but still uses a density model.
> > - **Density model**: The k-nearest neighbor approach used to estimate sparsity in the state representation embedding space is indeed a density model. We ask the reviewer for clarification on the statement "APT and similar methods leverage non-parametric k-nearest neighbor entropy estimators". To our understanding, the density estimator used (k-nearest neighbor) depends on representations produced by a parametric function.

---

> > > ### Comment · Reviewer_65e4 · 2024-11-25
> > >
> > > Thanks to the authors for their response. First, I appreciate the sharing of the code for the submission, which partially addresses my concerns raised in Weakness 2. I am also grateful for the clarification regarding the differences between $r_W$ and $r_{KL}$, which provides some intuition that partially addresses Weakness 3. In addition, I understand the nuances associated with quantifying state space coverage, which the reviewers have described in their response to Weakness 1.
> > >
> > > Nonetheless, several of my original concerns remain: (1) the "state coverage" performance metric upon which the experimental results presented in Table 1 rely is still insufficiently justified and no concrete justification or alternative performance metric has been provided (Weakness 1); (2) though the code has been released, no implementation description has been provided, making it difficult to fully understand and evaluate the experimental results (Weakness 2); (3) exactly how the theoretical results apply to the rest of the paper (Weakness 4) remains unclear. Finally, it appears that no revision has yet been provided (**please let me know if this is wrong or if the revision is subsequently uploaded**), so several of the items proposed in the author response remain incomplete. For these reasons, I maintain my original score.

---

### Meta-Review · Area_Chair_dHcV · 2024-12-19

**Metareview:**

This paper proposes an exploration strategy by maximizing the Shannon entropy of the state occupancy measure. This is achieved by maximizing a measure of divergence between successive state occupancy measures. The authors argue for the efficacy of their method by evaluating on a set of mazes and robotic manipulation and locomotion tasks.

The paper proposes a novel method that is theoretically motivated, for a problem that is of general interest to the RL community.

However, there is a general consent that the work is still lacking in its empirical evaluation. Further, there was a point raised on the bolded numbers in Table 1, which indeed seems to include performance gains which are not statistically significant.

As such, I will not be recommending acceptance of this work, and recommend the authors incorporate the reviewer feedback to further strengthen this submission.

**Additional Comments On Reviewer Discussion:**

The main concerns raise were with regards to empirical evaluations, as discussed above. There were a few points raised on the theoretical underpinnings and framework of RAMP, but most of these were successfully addressed by the authors.

---

### Decision · Program_Chairs · 2025-01-22

Reject